# Soft-consensual Federated Learning for Data Heterogeneity via Multiple Paths

**Sheng Huang[1], Lele Fu[1], Fanghua Ye[2], Tianchi Liao[1], Bowen Deng[1], Chuanfu Zhang[1], Chuan Chen[1]***

[1]Sun Yat-sen University, Guangzhou, China
[2]Tencent Inc., Shenzhen, China
{huangsh253, fulle, liaotch, dengbw3}@mail2.sysu.edu.cn
{zhangchf9, chenchuan}@mail.sysu.edu.cn
fanghua.ye.21@gmail.com

## Abstract

Federated learning enables collaborative training while preserving the privacy of all participants. However, the heterogeneity in data distribution across multiple training nodes poses significant challenges to the construction of federated models. Prior studies were dedicated to mitigating the effects of data heterogeneity by using global information as a blueprint and restricting the local update of the model for reaching a "hard consensus". But this practice makes it difficult to balance local and global information, and it neglects to negotiate amicably between local and global models to reach mutually agreeable results, called "soft consensus". In this paper, a multiple-path solving method is proposed to balance global and local features and combine these two feature preference paths to reach a soft consensus. Rather than relying on global information as the sole criterion, a negotiation process is employed to address the same objective by accommodating diverse feature preferences, thereby facilitating the discovery of a more plausible solution through multiple distinct pathways. Considering the overwhelming power of local features during local training, a swapping strategy is applied to weaken them to balance the solution paths. Moreover, to minimize the additional communication cost caused by the introduction of multiple paths, the solution of the task network is converted into data adaptation to reduce the amount of parameter transmission. Extensive experiments are conducted to demonstrate the advantages of the proposed method.

## 1 Introduction

Federated learning is a privacy-preserving distributed learning paradigm [1, 2, 3] that can coordinate several training nodes to jointly train a unified global model without exchanging raw data [4, 5, 6]. It is able to federate data from multiple parties to capture a diverse data distribution [7, 8]. As the significance of privacy has been increasingly emphasized [9, 10, 11], federated learning has been widely applied in fields [12, 13] with high privacy requirements, such as medical image processing [14, 15], recommendation systems [16, 17], Internet of Things [18] and so on.

As a classic federated learning method, FedAvg [19] establishes an important training paradigm. Numerous subsequent works [20, 21] are performed on the basis of this fundamental training paradigm. However, this training paradigm faces several challenges. Particularly, the problem of imbalanced data distribution [22] is widespread in the setting where federated learning is applied and can be attributed to the nature of distributed learning: joining multiple different data sources to participate in

---

*Corresponding author.

39th Conference on Neural Information Processing Systems (NeurIPS 2025).

the training process [23, 24, 25]. This would lead to disagreement among the optimization objectives of the participating nodes [26, 27].

The solution to the divergence of the optimization objectives at different nodes is usually to use global information to constrain the local update process [28, 29], thereby enforcing the local optimization objectives at nodes to be as close as possible to the global objective. This approach is referred to as *hard consensus*, as it imposes a rigid and immutable consensus across nodes. Typically, predefined indicators—such as constraints on parameter updates or controls on the distribution of generated representations—are employed to regulate the optimization process. However, these indicators are often indirect with respect to the target tasks, as no direct criteria are available for task-specific evaluation. Furthermore, they are coercive in nature, as the global norm remains fixed and unaltered, lacking flexibility or consultation. This rigid form of hard consensus may not always lead to optimal outcomes, as the global model may not perform better on local data than the existing client model.

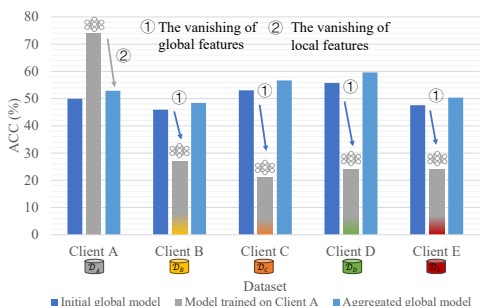

Figure 1: The classification accuracies achieved by the model on the training set of each client are reported at three stages: initialization, after being trained by client A, and after aggregation. The global model, which is trained on local data, will improve the classification accuracy on the current client, while the knowledge delivered by the other clients will dissipate, this is reflected in the decreased accuracies on the training set of other clients. Similarly, aggregating the local models results in the vanishing of the local features. Global and local features are in an antagonistic relationship.

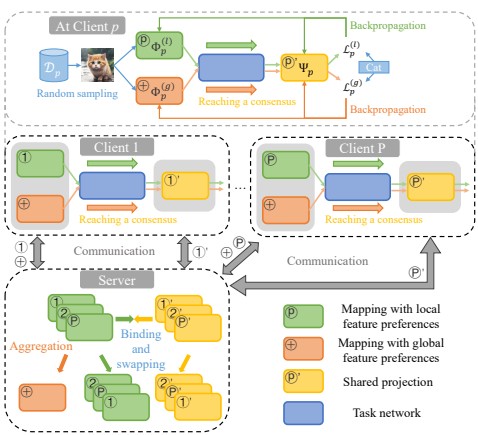

Figure 2: The architecture illustration of FedMP. On each client, the multiple paths solving is performed using mappings with different feature preferences and jointly optimizing the shared projection to reach a soft consensus between the global and local features. Clients only upload parameters other than the task network to save the communication costs. Multiple paths are balanced by an swapping strategy on the server.

Moreover, in federated learning, global and local features often compete with each other. A common approach to information exchange during each training round involves replacing the local model on the current node with the global model. However, this strategy can severely undermine the local features previously learned by the local model, leading to the dominance of global features. On the other hand, during the local training process, the local features have a greater advantage and take a dominant role, and continually influence the model under training. In this process, the dominance of global features would gradually vanish, the focus of the model turns to the local features, and the model would gradually forget what it has learned from the other clients. Global and local features alternately affect model training, these two features are in conflict, and are in an unbalanced and oscillating state. However, they both play a crucial role in the training of the final model. As shown in Figure 1, the local model has a high classification accuracy on the current client, however, it only receives its visible local data and therefore has a very high local feature preference with the forgetting of the global knowledge, which leads to the lower classification accuracy on the other clients. The global model, on the other hand, fuses the knowledge of various clients, has global feature preferences, and therefore is able to achieve relatively high performance on all clients. Nevertheless, this also hits the local features, which can be seen from the performance decrease on client A. To address this trade-off, it is essential to leverage both types of features and strike a balance between them, transforming their competitive relationship into a cooperative one.

For the *hard consensus* and the problem of local and global features *balancing* in federated learning, in this paper we propose a multiple paths solving approach, called **Fed**erated Learning Framework

with **M**ultiple **P**aths (FedMP), in order to balance the global and local features and prompt them to spontaneously generate a consensus, i.e., *soft consensus*. The architecture is shown in Figure 2. The proposed method integrates mappings with global and local feature preferences and federates their respective solution paths from distinct initialization points to achieve a consensus solution within a unified solution space. The shared solution space projection, being directly related to the target task, serves as a task-driven criterion. This projection is learnable and is jointly updated alongside the multiple solution paths, replacing the previous indirect and static hard consensus mechanism. Additionally, a swapping strategy is introduced to further balance the influence of local features, fostering a more harmonious cooperation between global and local features. Finally, the optimization objective is shifted from task-oriented to data-oriented, minimizing additional communication overhead in federated learning scenarios. Overall, the contributions of this paper are summarized as follows:

- A soft-consensual federated learning framework is constructed via the multiple paths that effectively combine local and global information, providing a platform for models with local features preferences and global features preferences to reach a soft consensus.

- The strengths of the paths are balanced among each other, and the swapping strategy is used to weaken the overly strong paths.

- To reduce the additional communication cost in federated learning scenarios, the solution target is shifted from the task to the data adapter for reducing the amount of parameter transmission.

- The effectiveness of the proposed method is demonstrated using extensive experiments, and exploratory experiments are conducted for the multiple paths approach.

## 2  Methodology

First, we present a formal description of federated learning and the motivation for the proposed method in Section 2.1. Then, in the following subsections, we describe the detailed implementation of the proposed method.

### 2.1  Problem Description and Overall Motivation

For a multiple device participating federated learning system, each participant is referred to as a client, and the $p$-th client has a dataset $\mathcal{D}_p$ that is a subset of the plenary data $\mathcal{D} = \{\mathcal{D}_1 \cup \cdots \cup \mathcal{D}_P\}$, where $P$ denotes the number of clients. The goal of federated learning is to combine the data of all participants to generate a global optimal model. Specifically, due to the privacy issue, individual clients cannot exchange the raw data. Therefore, each client has its own optimization objective $\min_{\theta_p} \mathcal{L}_p(\theta_p; \mathcal{D}_p)$ under the model parameters $\theta_p$, and the global objective is usually a weighted average of the local optimization objectives:

$$\min_{\hat{\theta}} \mathcal{L}\left(\hat{\theta}; \mathcal{D}\right) = \sum_{p=1}^{P} \frac{|\mathcal{D}_p|}{|\mathcal{D}|} \mathcal{L}_p\left(\hat{\theta}; \mathcal{D}_p\right), \tag{1}$$

where $|\cdot|$ denotes the number of elements in the set and $\hat{\theta}$ means the global model parameters. During the training process, periodic information exchange is necessary to tune the training procedure of each client, preventing overfitting to the local data distribution and ensuring the model remains suitable for the global data distribution. The server, which acts as the regulator of the federated learning system, typically broadcasts the current global model information at each communication round. This helps adjust the local models by providing a fresh optimization starting point. After the client completes the current training, the server accepts updates from the clients in order to collect the information provided by all participants, and aggregates them into the global model. The information delivered here is usually the parameters or the gradient of the model, and in some cases also contains other auxiliary information.

**Motivation**   Due to the existence of non-IID data, global and local models typically exhibit conflicting feature preferences, and simply distributing or aggregating models can be detrimental to both global and local features. The global model, as a product of the aggregation from conflicting features,

contains information that may not all be absolutely correct. Therefore, the mechanism of using global information as a hard criterion to drive local models in order to create a hard consensus between local features and global features has potential for improvement. Moreover, using global information as a reference often lacks consideration of the target task, and a task-driven coordination approach is worth investigating. How to harmonize global and local information, preserve as complete information as possible with balance among them, and make them produce mutually agreeable results on the target task to reach a soft consensus is the focus of this work.

## 2.2 Multiple Paths High Confidence Solving with Information Relics

A common approach to information transfer in federated learning is model transmission. Through the use of global model sending and replacing the local models, so as to use client specific data to adjust the mean of the information provided by the previous clients involved in the training. And global information may also be used as the criterion to control local updates. This is a very intuitive way, but lacks consideration of the balance between global and local information, and the information relics of local features. We believe that a softer approach with more comprehensive reference information should be used, and reach a soft consensus on global and local models. Therefore, the proposed multiple paths solving approach is described below.

For a typical neural network, there is a stream of data directed from the inputs to the outputs, which maps the input space to the output space. We have the following definition:

**Definition 1.** *(Solution path). It is expected that there is a mapping $f(\cdot)$ which makes the data $x$ map to the target $y = f(x)$. Fitting this mapping using a neural network $g(\cdot)$, defines the process of optimizing $g(x)$ to $g^*(x)$ as the solution path. Here, $\|g^*(x) - f(x)\| \leq \epsilon$, and $\epsilon$ is the error.*

**Definition 2.** *(Different solution paths). For mapping $f(\cdot)$, the optimization of neural network $g_1(\cdot)$ to $g_1^*(\cdot)$ and neural network $g_2(\cdot)$ to $g_2^*(\cdot)$ will be the different solution paths. Here, $\|g_1^*(x) - f(x)\| \leq \epsilon_1$, $\|g_2^*(x) - f(x)\| \leq \epsilon_2$, where $\epsilon_1$ and $\epsilon_2$ are the errors.*

From the above definitions, it follows that the model trained locally encapsulates knowledge refined from the data of a single client, resulting in one solution path, while the global model distributed by the server, which incorporates knowledge refined from the data of all participants, leads to another solution path. Relying on a single solution path often results in biased information and increases the likelihood of the neural network becoming trapped in local optima. To address this issue, information from multiple solution paths should be utilized simultaneously. This approach ensures the aggregation of diverse information sources, preventing the overwriting and forgetting of knowledge associated with a single information channel. Furthermore, the concurrent use of multiple solution paths offers an alternative to the traditional practice of using global information as a rigid criterion, promoting a more flexible and adaptive learning framework. For the mapping from input to output on client $p$, constructed using the multiple paths way, it can be trained by the following loss function:

$$\mathcal{L}_p = \frac{1}{|\mathcal{D}_p|} \sum_{i=1}^{|\mathcal{D}_p|} \left[ \mathcal{L}_p^{(l)}(\Psi_p(\Phi_p^{(l)}(\mathbf{X}_p^i))) + \gamma \mathcal{L}_p^{(g)}(\Psi_p(\Phi_p^{(g)}(\mathbf{X}_p^i))) \right], \tag{2}$$

where $\mathbf{X}_p^i \in \mathcal{D}_p$ is the $i$-th data sample of the $p$-th client, $\mathcal{L}_p^{(l)}$ and $\mathcal{L}_p^{(g)}$ denote the loss functions for the local and global solution paths, respectively, $\gamma$ is the path balancing hyperparameter, $\Phi_p^{(l)}(\cdot)$ and $\Phi_p^{(g)}(\cdot)$ represent the mappings under two different solution paths, which map the input data to the latent space, and $\Psi_p(\cdot)$ denotes the projection that maps the data in the latent space to the eventual output space. Note that here $\Psi_p(\cdot)$ is shared by all solution paths. The shared projection serves to make the objectives of all multiple solution paths located in the same solution space, and to ensure that all solution paths have the same target by employing the same loss function. Solving for an identical objective from different paths leads to a solution with very high confidence. This is based on a highly intuitive assumption: if different approaches yield the same or very similar results, then such results can be considered highly reliable. Therefore, by using this loss function, the mappings are solved from both a solution path biased towards local features and a solution path biased towards the global features, which gently collaborates the global and local information, and prevents overwriting and forgetting of the information. The soft consensus generated by the global and local information is materialized in a shared projection, and the mutually agreeable results are reached through their respective solution paths. The shared projection guarantees that the space of solutions is mapped in a

way that is more suitable for reaching consensus and is directly related to the label space, which is task-driven. **Further discussion on multiple paths can be found in the Supplementary Material.**

## 2.3   Local Preference Swapping Strategy

From the multiple paths solving process described in Section 2.2, the following sequences of initial models in each communication round are given at the $p$-th client:

$$\mathbf{\Psi}_p = \{\psi_p^1, \psi_p^2, \cdots, \psi_p^r, \cdots, \psi_p^R\}, \tag{3}$$

$$\mathbf{\Phi}_p^{(l)} = \{\phi_p^1, \phi_p^2, \cdots, \phi_p^r, \cdots, \phi_p^R\}, \tag{4}$$

$$\mathbf{\Phi}_p^{(g)} = \{\phi^1, \phi^2, \cdots, \phi^r, \cdots, \phi^R\}, \tag{5}$$

where $\phi_p^r$ denotes the model used for approximate mapping $\Phi_p^{(l)}(\cdot)$ in $r$-th communication round at $p$-th client, and $\psi_p$ is the model used for approximate the projection $\Psi_p(\cdot)$. Here, $\phi^r = \sum_{p=1}^{P} \frac{|\mathcal{D}_p|}{|\mathcal{D}|} \phi_p^{r-1}$. When $r = 0$, the whole system is in initialization state, so that $\phi_p^1 = \phi_p^0$.

In federated learning, restriction on access to data is a key factor affecting model learning. Local models trained too much on the current dataset would overfit the visible distribution and have difficulty in adapting to the global distribution even if there is the multiple paths method to correct for it. Therefore, the multiple paths network for federated learning needs to be modified in terms of training strategy.

It is easy to find that all conditions required for the multiple paths method can be summarized as, a solution path with a preference for local features and a solution path with a preference for global features, and to optimize on both solution paths simultaneously. Therefore, it is only necessary to provide solution paths under other mapping preferences which are different from the global features to satisfy the requirements for the multiple paths formulation. A local preference swapping strategy can be used to avoid the local model overfitting problem by replacing the local model in the current client with random local model of other clients downloaded from the server at the beginning of each communication round. For the $p$-th client, we can convert the sequence of initial models Eq. (4) and Eq. (3) to the following form:

$$\mathbf{\Phi}_p^{(l)} = \{\phi_p^1, \phi_{p^{2'}}^2, \cdots, \phi_{p^{r'}}^r, \cdots, \phi_{p^{R'}}^R\}, \tag{6}$$

$$\mathbf{\Psi}_p^{(l)} = \{\psi_p^1, \psi_{p^{2'}}^2, \cdots, \psi_{p^{r'}}^r, \cdots, \psi_{p^{R'}}^R\}, \tag{7}$$

where $p^{r'}$ is a random sample of the client indexes that participated in the previous round of training. In this way, each local model of the participating clients is updated after each round of communication, so that the multiple paths method obtains the optimized origin of the solution path that is preferred to the local features, and avoids the overfitting because the models do not stay at one client all the time. Additionally, the swapping strategy enables each model to accept a wider range of inputs, which improves the generalization ability of the model.

## 2.4   Multiple Paths Data Adaptation

In federated learning, communication cost is a topic worth considering. Compared to the classical federated learning methods such as FedAvg, multiple paths solving would require additional network communication cost. Specifically, at each communication round, the contents to be exchanged include the mapping with local feature preferences and the mapping with global feature preferences. Although the additional communication cost of multiple paths solving is not too large compared to methods such as SCAFFOLD that require uploading a large amount of auxiliary information, we still consider further reducing the additional communication cost associated with multiple paths solving.

Originally, we employed the multiple paths solving directly to the full task network, which would have transmitted the network parameters under different multiple solution paths. The communications volume is positively correlated with the number of current task network parameters. A mind shift can be made to shift the target of multiple paths solving from the task network to the adaptation of the data, and to relax the optimization target for the full task network, relocating the more stringent optimization target to the adaptation of the data. The scale of the data adaptation network can be small, but the solution objective is precise, and the exchange of information among the individual

clients only occurs over this part of the network parameters, which greatly reduces the communication cost during each communication round. Therefore, a modification of Eq. (2) yields the following loss function:

$$\mathcal{L}_p = \frac{1}{|\mathcal{D}_p|} \sum_{i=1}^{|\mathcal{D}_p|} \left[ \mathcal{L}_p^{(l)}(\Psi_p(\Omega_p(\tau_p^{(l),i}))) + \gamma \mathcal{L}_p^{(g)}(\Psi_p(\Omega_p(\tau_p^{(g),i}))) \right], \tag{8}$$

$$s.t. \ \tau_p^{(l),i} = \Phi_p^{(l)}(\mathbf{X}_p^i), \ \tau_p^{(g),i} = \Phi_p^{(g)}(\mathbf{X}_p^i).$$

Here, $\Omega_p(\cdot)$ denotes the full task network on the $p$-th client, $\tau_p^{(l),i}$ and $\tau_p^{(g),i}$ represent the adapted data generated from the two mappings on different solution paths, respectively. $\Omega_p(\cdot)$ remains constant during the multiple paths solving process, in which the fitting of the network to the task is transformed into the fitting of the data to the network, i.e., the data attempts with some rules to mimic the patterns that can be recognized by the network. This is a shift in the goal of the solution and this shift needs to be solved by multiple paths solving in order to obtain a more accurate solution. For the task network, extracting the hidden deep information in the data requires a very complex network with huge amount of network parameters to be updated to suit the task. While by simply adapting the data to convert them into the features which are required for the task network, the size of the network required can be tiny since this does not require the decomposition of the deep information in the data. This may intuitively diminish some of the performance, but we are surprised to find that in conjunction with the multiple paths solving method, the expected performance degradation is not very noticeable, which can be verified by the subsequent experimental results.

## 2.5 Training Details

Since the multiple paths solving method requires different solution paths for the solving process, and we want all the solution paths employed are reliable. Hence, in the early stage of training, we use a few global communication rounds employing the same training strategy as FedAvg in order to obtain a more reliable solution path with global feature preferences. At this stage, the model at $p$-th client is trained using the following loss function:

$$\mathcal{L}_p = \frac{1}{|\mathcal{D}_p|} \sum_{i=1}^{|\mathcal{D}_p|} \mathcal{L}_p(\Psi_p(\Omega_p(\tau_p^i))), \quad s.t. \ \tau_p^i = \Phi_p(\mathbf{X}_p^i). \tag{9}$$

And, after each communication stage, the same global model is used to replace the client model for each client. After completing the initial training process, the parameters of the task network $\Omega_p(\cdot)$ on $p$-th client are not changed and the parameters of these networks are no longer exchanged in the communication. Then, for the $p$-th client, the multiple paths solving process is performed using Eq. (8), and the local preference swapping strategy proposed in Section 2.3 is used to allow each client to obtain a more reliable solving path with local feature preferences. Additionally, considering that the performing of the swapping strategy after several communication rounds may result in the mappings preferring local features overly preferring local features, we use the mappings with the global feature preferences every $B$ rounds for replacement in order to balance the influence of the local features. Since the models are only exposed to local data during training, only the mappings with preferences for local features require additional balancing. The detailed algorithm is summarized in Algorithm 1 (See Supplementary Material).

## 3 Theoretical Analysis

In this section, we provide a convergence analysis of the proposed FedMP. First, some assumptions are introduced to help complete the following theoretical analysis.

**Assumption 1.** *For any $p \in [P]$, local loss function for local solution path $\mathcal{L}_p^{(l)}$ and local loss function for global solution path $\mathcal{L}_p^{(g)}$ are L-smooth with respect to $\Theta$. For $\forall \ \Theta \ and \ \Theta'$, the following inequalities hold:*

$$\|\nabla \mathcal{L}_p^{(l)}(\Theta) - \nabla \mathcal{L}_p^{(l)}(\Theta')\| \leq L_1 \|\Theta - \Theta'\|, \tag{10}$$

$$\|\nabla \mathcal{L}_p^{(g)}(\Theta) - \nabla \mathcal{L}_p^{(g)}(\Theta')\| \leq L_2 \|\Theta - \Theta'\|, \tag{11}$$

*where $L_1$ and $L_2$ are Lipschitz constants.*

**Assumption 2.** *The upper bound on the variances of the local gradient to the aggregated mean can be given as follows*

$$\frac{1}{P}\sum_{p=1}^{P}\|\nabla\mathcal{L}_p^{(l)}(\Theta) - \nabla\mathcal{L}^{(l)}(\Theta)\|^2 \leq \delta_L^2, \tag{12}$$

$$\frac{1}{P}\sum_{p=1}^{P}\|\nabla\mathcal{L}_p^{(g)}(\Theta) - \nabla\mathcal{L}^{(g)}(\Theta)\|^2 \leq \delta_G^2, \tag{13}$$

*where $\delta_L$ and $\delta_G$ are the constants.*

Thus, Theorem 1 can be derived:

**Theorem 1.** *Suppose Assumption 1 and 2 hold, the convergence property of the proposed method can be described by*

$$\frac{1}{R}\sum_{r=0}^{R-1}\mathbb{E}\left[\|\nabla\mathcal{L}(\Theta^r)\|^2\right] \leq \epsilon, \tag{14}$$

*where $\epsilon = \frac{\mathbb{E}[\mathcal{L}(\Theta^0) - \mathcal{L}(\Theta^R)]}{\mathcal{H}R} + \mathcal{S}$, and*

$$\mathcal{H} = \frac{\eta EB}{2}\left[1 - 2\eta EB(L_1 + \gamma L_2) - 32\eta^2 E^2 B^2(L_1 + \gamma L_2)^2 \right. \\ \left. -64\eta^3 E^3 B^3(L_1 + \gamma L_2)^3\right], \tag{15}$$

$$\mathcal{S} = \left[\frac{\eta EB}{2} + \eta^2 E^2 B^2(L_1 + \gamma L_2)\right]\left[64\eta^2 E^2 B^2(L_1 + \gamma L_2)^2 + \frac{4}{K}\right] \\ \frac{P-K}{P-1}(\delta_L^2 + \gamma^2\delta_G^2). \tag{16}$$

From Theorem Theorem 1 we can know that the proposed FedMP can reach the convergence with appropriate hyperparameters choices. See Algorithm 1 for definitions of $R$, $E$, $B$, and $K$.

*Proof.* Please see Supplementary Material for detailed proof. □

## 4 Experiments

### 4.1 Experimental Setup

**Datasets:** We conduct main experiments with the proposed method using classification tasks on three datasets, which are CIFAR-10 [30], CIFAR-100 [30] and Flowers102 [31]. CIFAR-10 and CIFAR-100 have 10 and 100 classes, respectively, and Flowers102 has 102 classes. For CIFAR-10 and CIFAR-100 datasets, we use a train set consisting of 50,000 samples and a test set consisting of 10,000 samples. For Flowers102 dataset, we use a train set consisting of 6,149 samples and a test set consisting of 1,020 samples. We partition the train sets using the Dirichlet distribution with hyperparameter $\alpha \in \{0.3, 0.5, 1.0\}$ to simulate the scenarios with the heterogeneous data distribution, where the smaller $\alpha$ is, the more unbalanced the data distribution among clients. The distributions of the train set and other settings can be found in Supplementary Material.

**Baselines:** In order to compare the results, we also run nine SOTA federated learning methods under the same datasets setup, including FedAvg [19], FedProx [20], SCAFFOLD [21], FedNova [32], FedDF [33], MOON [28], FedASAM [34], FedPVR [29] and FedUCS [12].

Experimental results are shown in the following and the Supplementary Material.

### 4.2 Experimental Results

#### 4.2.1 Experiment of Multiple Paths

We first validate the effectiveness of multiple path solving in centralized learning. We use the three test datasets as the training data for the classification task. The MLP module is used as the shared

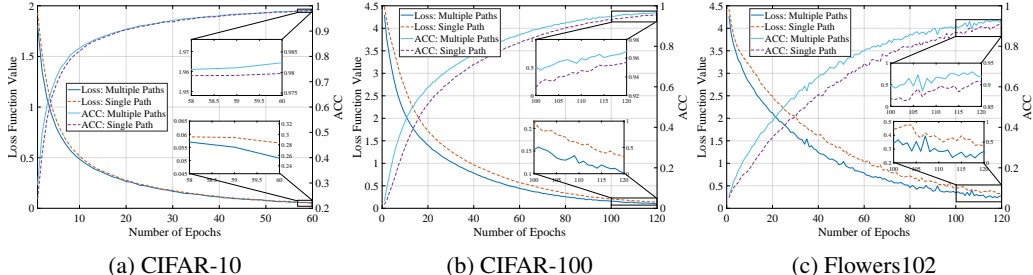

(a) CIFAR-10         (b) CIFAR-100         (c) Flowers102

Figure 3: The curves of loss function values and classification accuracies during training for single path solving and multiple paths solving. Multiple paths solving can make the model find a fitter solution faster and get a better performance.

projection and two ResNet-18 modules are used as the multiple solution paths. Since the definition of global and local features does not exist in centralized learning, we use two different initialization methods as two different solution paths. One of the solution paths is taken out individually and used the same training setup for the task network under the single path, and this network is also used as the task network in the multiple path setup as well. The loss and training accuracies curves that vary with epochs during training are demonstrated in Figure 3. From the figure, it can be seen that the task network in multiple paths can obtain faster learning speed and better final learning results than the single-path network without changing the network structure and the optimization method settings. The multiple paths solving method can combine the multiple solution paths to form a soft consensus, and avoiding the limitations of single path solving. The knowledge learned by different paths can effectively provide assistance among paths, and prevent a single path to be trapped in a local optimal solution. Therefore, we can think that the multiple path solving method can effectively help the network to obtain more comprehensive and reliable information and improve the training performance by reaching a consensus among multiple paths. The experimental results demonstrate that the proposed multiple paths solving method is effective. And this effectiveness is the strong motivation for us to apply this method in federated learning.

### 4.2.2 Main Results

Here, we show the classification accuracy results of the proposed method with eight other comparative methods on three datasets in Table 1. It can be seen that the proposed method is able to better optimal performance in most of the dataset settings. The proposed method is able to achieve a greater advantage under the scenarios of higher data heterogeneity or the more difficult classification tasks, which we attribute to the fact that the multiple paths solving approach provides more comprehensive auxiliary information and replaces the hard consensus by forming a soft consensus between global and local features, and reversing the information that may be biased in the global information. With the multiple paths solving, the global and local information, which are opposing to each other, are converted into a cooperative relationship in order to perform federated learning more comprehensively from different perspectives.

Table 1: The results of three datasets for all methods. **Bold** and underlined results are the best and the second best. "↑" indicates accuracy improvement over FedAvg.

| Method | CIFAR-10 | | | CIFAR-100 | | | Flowers102 | | |
|---|---|---|---|---|---|---|---|---|---|
| | $\alpha$=0.3 | $\alpha$=0.5 | $\alpha$=1.0 | $\alpha$=0.3 | $\alpha$=0.5 | $\alpha$=1.0 | $\alpha$=0.3 | $\alpha$=0.5 | $\alpha$=1.0 |
| FedAvg | 80.66 | 83.34 | 87.46 | 57.25 | 59.13 | 61.34 | 41.86 | 43.92 | 45.49 |
| FedProx | 81.16 | 83.67 | 88.04 | 56.37 | 57.77 | 62.36 | 34.90 | 37.35 | 39.22 |
| SCAFFOLD | 80.60 | 86.08 | 87.66 | 58.65 | 60.96 | 61.81 | **45.98**↑4.12 | 46.37 | 48.92 |
| FedNova | 83.41 | 84.93 | 86.41 | 55.46 | 55.02 | 56.67 | 41.08 | 40.69 | 43.73 |
| FedDF | 78.70 | 76.95 | 88.65↑1.19 | 52.82 | 52.46 | 59.32 | 22.96 | 24.90 | 24.09 |
| MOON | 82.47 | 85.46 | 87.83 | 57.52 | 59.30 | 62.85 | 41.08 | 43.33 | 45.88 |
| FedASAM | 68.98 | 71.59 | 74.44 | 46.84 | 47.91 | 48.52 | 30.20 | 30.29 | 30.59 |
| FedPVR | 84.65↑3.99 | 85.73 | 87.86 | 59.32 | 60.89 | 62.75 | 43.04 | 44.41 | 45.88 |
| FedUCS | 83.61 | 87.79↑4.45 | **89.43**↑1.97 | **62.19**↑4.94 | **63.24**↑4.11 | **64.28**↑2.94 | 43.72 | 49.22↑5.30 | 51.18↑5.69 |
| FedMP | **85.37**↑4.71 | **88.96**↑5.62 | 88.31 | 61.43↑4.18 | 62.04↑2.91 | 62.95↑1.61 | 43.73↑1.87 | **54.41**↑10.49 | **53.43**↑7.94 |

### 4.2.3 Ablation Study

In order to validate whether the various proposed enhancement strategies are effective in the federated learning scenario applied to heterogeneous data, we perform the ablation study and demonstrate

the results in Table 2. Specifically, for the two main strategies we proposed: multiple paths solving and swapping strategy, we make them available or not and train the frameworks constructed by all possible configurations under the same dataset setup, yielding the results in Table 2. Taking the first row of data in the table as the baseline, from the other data in the table we can see that when using only the multiple paths approach, since one of the paths uses the local models without swapping, the model predictably fits more strongly to the local data, and leading to the overfitting for the local features. Therefore, this path will affect the learning of global features, and even with the assistance of global features, it is still unable to form a more correct consensus, so that the performance decreases compared to the baseline. The model that only employs the swapping strategy is unable to obtain the update of global features in time, and thus cannot obtain a stable improvement for the learning of the global model, which is rather a negative impact on the global model at the CIFAR-10 dataset with the heterogeneous parameters of 0.3 and 0.5. In other settings, there are boosts, but the magnitudes are more limited. In contrast, when both proposed strategies are applied to the model, the more significant and stable boosts are obtained in most settings.

In order to demonstrate more intuitively the effectiveness of the proposed multiple paths solving approach in a federated system, the results of the classification accuracy achieved on the training data of each participating client for both local models and global models are presented in Figure 4. From the figure, it can be seen that when using the multiple paths solving approach for training, it is usually possible to achieve a more excellent performance, both for the local and global models. When using a single-path architecture, the absence of a global path eliminates the need to account for global features, allowing the model to better fit local data. However, this also limits the single-path approach's ability to generalize to data from other clients. Additional experimental results are provided in the supplementary material.

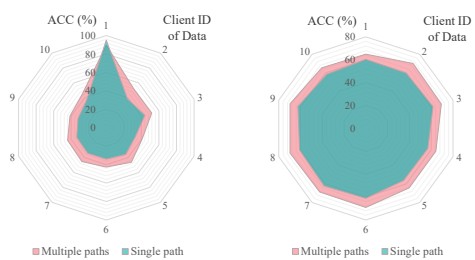

(a) Local model of Client 1    (b) Global model

Figure 4: Classification accuracies of the two models on the data in different clients. The local model trained by the multiple paths is able to achieve better performance on the data of other clients, and it can increase the performance of the aggregated global model.

Table 2: Ablation study of FedMP on three datasets. **Bold** results are the best.

| Multiple Paths | Swapping Strategy | CIFAR-10 | | | CIFAR-100 | | | Flowers102 | | |
|---|---|---|---|---|---|---|---|---|---|---|
| | | $\alpha$=0.3 | $\alpha$=0.5 | $\alpha$=1.0 | $\alpha$=0.3 | $\alpha$=0.5 | $\alpha$=1.0 | $\alpha$=0.3 | $\alpha$=0.5 | $\alpha$=1.0 |
| × | × | 83.66 | 85.32 | 85.45 | 60.40 | 61.06 | 62.29 | 38.33 | 52.32 | 49.80 |
| × | ✓ | 83.16 | 84.97 | 85.50 | 60.56 | 61.31 | 62.79 | 43.04 | 54.12 | **54.12** |
| ✓ | × | 82.96 | 81.47 | 82.82 | 56.97 | 58.26 | 59.71 | 38.63 | 51.86 | 49.41 |
| ✓ | ✓ | **85.37** | **88.96** | **88.31** | **61.43** | **62.04** | **62.95** | **43.73** | **54.41** | 53.43 |

## 4.3 FedMP with Full Multiple Paths

Here, instead of using the communication cost reduction method proposed in Section 2.4, we use a multiple paths solving method for the complete task network. We use the complete task network here as the mapping with global feature preferences and the mapping with local feature preferences, respectively, and use the same shared projection as in the experiments to motivate a soft consensus. We show the classification accuracy results in Figure 5. It can be seen that the results obtained are further improved after using the full multiple paths (FedMP-F). However, the penalty for this enhancement would be a higher communication cost, and we believe that the small amount of performance degradation associated with the reduction in communication cost is acceptable. Therefore, there is flexibility to trade-off between the two approaches depending on the requirements.

## 4.4 Data Privacy

As a privacy-preserving machine learning paradigm, federated learning avoids the transmission of raw data, thereby providing a certain level of privacy protection for participants. However, it remains possible to infer sensitive information about the original data from model parameters updates. To evaluate the adaptability of additional privacy-enhancing strategies, we conduct further experiments. Differential Privacy (DP), a well-established protection mechanism, is employed by injecting noise

into the transmitted model parameter updates. We use a backbone network consisting of two CNN layers and two fully connected layers, and apply DP with a noise multiplier of 1.0 on the CIFAR-10 dataset. The results are shown in the Figure 6. As the table indicates, even with the addition of privacy-preserving mechanisms, the proposed method still maintains a relative performance advantage. This demonstrates that FedMP can be extended with such strategies to meet stronger privacy protection requirements.

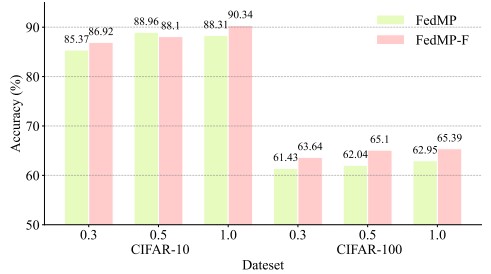

Figure 5: A comparison of global model performance with and without the complete multi-path network. The results show that adopting the full network leads to improved performance of the global model.

Figure 6: Comparison of global model performance with and without differential privacy under different data settings. The results show that FedMP maintains its advantage even when privacy-preserving mechanisms are applied.

## 4.5 Parameter Sensitivity Analysis

This subsection explore the impact of path balancing parameter $\gamma$ on the multiple paths solution effect. On the CIFAR-100 dataset, by setting different path balancing parameters values and recording the test accuracy, the results are shown in Table 3. As shown in the table, when the path balancing parameter is appropriately tuned, the multiple paths solving method achieves relatively stable and excellent performance. However, if the parameter is set too high or too low, it adversely impacts the final performance. This negative effect is particularly pronounced when the global path dominates. We speculate that this is because the global path is derived from the aggregation of local solution paths. Assigning excessive weight to the global path disrupts the optimization of local paths. In turn, poorly optimized local paths negatively influence the aggregated global path, amplifying the adverse effects in subsequent multiple paths updates. Thus, balancing the two paths is critical. An appropriately balancing parameter enhances the effectiveness of the multiple paths solving method, allowing it to achieve better performance.

Table 3: The results of classification accuracy under different path balance parameter values on the CIFAR-100 dataset. **Bold** and underlined results are the best and the second best.

| $\gamma$ | 0.01 | 0.05 | 0.1 | 0.5 | 1 | 5 | 10 | 50 | 100 |
|---|---|---|---|---|---|---|---|---|---|
| $\alpha$=0.3 | 60.76 | 60.46 | **61.43** | 60.17 | 59.22 | 2.57 | 23.92 | 1.00 | 1.00 |
| $\alpha$=0.5 | 61.35 | 61.13 | **62.04** | 60.91 | 60.18 | 9.19 | 16.33 | 1.00 | 1.00 |
| $\alpha$=1.0 | **63.10** | 62.94 | 62.95 | 62.89 | 62.76 | 19.21 | 14.50 | 1.00 | 1.00 |

## 5 Conclusion

In this paper, we propose a federated learning framework with multiple paths for balancing global and local features, and motivating them to reach a soft consensus. By introducing a multiple paths solving method, mappings with global and local feature preferences are federated to provide multiple solution paths for reaching a soft consensus among them. Since the balance of multiple solution paths is important, additional swapping strategy is introduced to equalize the strength of paths. Switching the objective of solving from the task network to the adaptation of data reduces the communication cost. We have performed various experiments, including exploring the effectiveness of multiple paths, to demonstrate that the proposed federated learning framework with multiple paths is effective. In the future, we will explore better ways of constructing multiple paths to achieve better performance.

## Contribution Statement

Sheng Huang and Lele Fu contributed equally to this work.

## Acknowledgments

The research is supported by the National Key R&D Program of China (2023YFB2703700), the National Natural Science Foundation of China (62176269).

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

# A Related Work

Federated learning needs to be performed without exchanging raw data in order to protect the privacy of all participants [35, 36, 37]. This paper focuses on federated learning in non-IID scenarios, which is the case where the distribution of data is heterogeneous on distributed clients. As a pioneering work on federated learning, FedAvg [19] joins distributed data for training by communicating model updates to achieve the target that several nodes co-train the global model. However, it leads to performance degradation in scenarios where data distributions have a large variance, which is used to process non-IID data [38, 39]. There are a number of works [40, 41, 42, 43, 44] that have been proposed to tackle the impact of data heterogeneity on federated learning. SCAFFOLD [21] introduces a method for controlling local updates by utilizing variance reduction techniques on client-side updates to overcome the effects of client drift. Similar ideas for controlling client-side updates have been adopted by various works, e.g., FedProx [20] controls the divergence between the local and global models by proposing a proximal regularization term to minimize the difference of the local model and the global model, FedDyn [45] employs a similar technique. MOON [28] makes each local update process not to deviate too far from the global model through contrastive learning. FedPVR [29] corrects for client updates by variance control at the end of the local models. Above and other works [46, 47, 48] control the direction of client-side optimization by enshrining global information as the norm, which produces a hard consensus that limits the preservation of local features that may be beneficial to the final model. And, there is also a lack of consideration of the balance between the global and local features.

# B Discussions

## B.1 More Discussion of Multiple Path

Under the concept of solution path, an intuitive illustration in Figure 7 shows the difference between the proposed multiple paths method and other ways of training neural networks. Many federated learning methods use the following approaches in the process of local updating. For a classical neural network as shown in Figure 7a, it is necessary to accept an input and make the neural network generate an output that is as close as possible to the expected value with the guidance of the loss function. There is only one solution path in this training approach, which is to adjust the parameters of the network to a more optimal value based on the current inputs and outputs of the network. For the network with contrastive learning as shown in Figure 7b, a group of inputs need to be accepted and positive and negative pairs need to be defined in order to

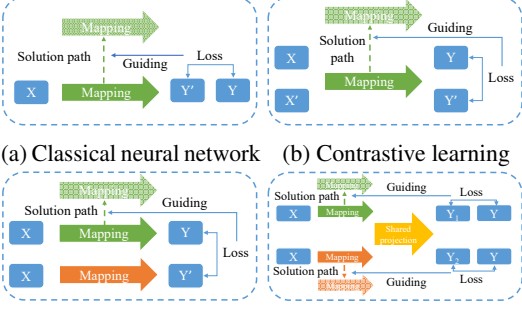

(a) Classical neural network    (b) Contrastive learning

(c) Knowledge distillation    (d) Multiple path solving

Figure 7: An illustration of the difference between the proposed multiple paths method and other ways of training neural networks.

bring the representations of the positive pairs closer and pull the representations of the negative pairs farther away. The solution path in this approach can be considered as the guidance for optimizing the network by data similarity, and there is still only one solution path. For the knowledge distillation as shown in Figure 7c, it is required to use a pair of networks, which includes a student network and a teacher network, that receive the same inputs and use the output of the teacher network as a criterion for tuning the parameters of the student network. This approach employs the information provided by the teacher network as the additional knowledge for the solution path of the network parameters optimizing, which is similar to contrastive learning, also with only one solution path, although it looks like two. The proposed multiple paths solving as shown in Figure 7d is similar to the knowledge distillation method in that two different networks receive the same inputs, however, the difference is that the two networks with different paths are solving for the same objective at the same time. By sharing the projection $\Psi(\cdot)$ and the same loss function, the two networks are ensured to be in a unified solution space in the latent space, which thus ensures that they have the same solution target. The multiple solution paths can be different mappings or different mapping preferences. This approach can be described as *All roads lead to Rome*. We use different mapping preferences as two

different solution paths, because there are two completely different feature tendencies in federated learning, i.e., global features and local features. Therefore, multiple paths solving is very suitable for federated learning.

## B.2 Discussion about the Swapping Strategy

In this paper, we propose a multiple paths solving approach for federated learning, and a swapping strategy is used in order to obtain more reliable solution paths. This subsection discusses the performance risk and privacy risk of the swapping strategy.

**Performance risk**   The swapping strategy requires additional model parameters processing and introduces additional communication costs. This increases the computational and communication consumption of the federated learning system. Therefore, to mitigate this performance risk, we propose a multiple paths data adaptation mechanism in the paper. This mechanism converts the solving of the task network into the adaptation of data, reducing the number of model parameters that need to be exchanged.

**Privacy risk**   The risk of privacy leakage that may result from the swapping strategy is explained here. Other clients would receive a model of some client from the server when the swap occurs, which may raise the privacy leakage of the original training data. Moreover, during the transmission of the models, there is also the possibility of information leakage due to unreliable communication links. But, there are still possible solutions to the above privacy risks. Since the execution party of the swapping strategy is the server, for the models that are distributed, the each client receives a model that is usually from a random client, and the model has been trained by several clients before it is distributed to the current client, which make it difficult to infer the gradient and personalization information of a specific client. Hence, it increases the difficulty of the privacy attack. In addition, each round of model assignment is randomized, which leads to difficulty in tracking the flow of specific models, which further protects client privacy. By applying differential privacy technique [49, 50] to the transmitted data, the protection of privacy can be obtained even further.

## C   Algorithm Details

We present the complete algorithmic workflow of the proposed FedMP in Algorithm 1. Algorithm 2 provides the detailed client-side optimization workflow implemented in Algorithm 1. And, we summarize all symbols used in the paper in Table 4.

| Symbol | Meaning |
|---|---|
| $\mathcal{D}_p$ | The dataset of the $p$-th client, which is a subset of the global data |
| $\|\cdot\|$ | The number of elements in a set |
| $\hat{\theta}$ | Global model parameters |
| $\theta_p$ | Model parameters on the $p$-th client |
| $\mathbf{X}_p^i$ | The $i$-th data sample of the $p$-th client |
| $\mathcal{L}_p^{(l)}$ | The loss function for the local solution path on the $p$-th client |
| $\mathcal{L}_p^{(g)}$ | The loss function for the global solution path on the $p$-th client |
| $\gamma$ | The path balancing hyperparameter, used to balance the influence between local and global paths |
| $\Phi_p^{(l)}(\cdot)$ | The mapping of the local solution path on the $p$-th client |
| $\Phi_p^{(g)}(\cdot)$ | The mapping of the global solution path on the $p$-th client |
| $\Psi_p(\cdot)$ | The shared projection on the $p$-th client |
| $\phi_p^r$ | The model used by the $p$-th client in the $r$-th communication round to approximate $\Phi_p^{(l)}(\cdot)$ |
| $\psi_p$ | The model used by the $p$-th client to approximate the projection $\Psi_p(\cdot)$ |
| $p^{r'}$ | A random sample of client indices that participated in the previous round of training |
| $\Omega_p(\cdot)$ | The task network on the $p$-th client |
| $\tau_p^{(l),i}$ | The adapted data generated on the $p$-th client of the $i$-th data sample through the local path mapping |
| $\tau_p^{(g),i}$ | The adapted data generated on the $p$-th client of the $i$-th data sample through the global path mapping |

Table 4: List of symbols and their meanings.

---

**Algorithm 1** Federated Learning Framework with Multiple Paths (FedMP)

---

**Input**: The number of communication round $R$, The number of initial training round $R_{it}$, The number of balance round $B$, the number of local epochs $E$, hyperparameter $\gamma$, the learning rate $\eta$ and the number of selected clients $K$ per round.

**Output**: The global model $\Theta = \{\Psi, \Omega, \Phi\}$.

1: **Server**:
2:     Initialize the weights $\psi^0$, $\omega^0$ and $\phi^0$ of $\Psi$, $\Omega$ and $\Phi$, respectively;
3:     **for** $r = 1 \rightarrow R$ **do**
4:         $\mathcal{C} \leftarrow$ (Randomly selects $K$ clients);
5:         **for** $p$-th client $\in \mathcal{C}$ **do**
6:             **if** $r \leq R_{it}$ **then**
7:                 $\psi_p^r, \omega_p^r, \phi_p^r \leftarrow$ **Client**$(p, r, \psi^{r-1}, \omega^{r-1}, \phi^{r-1})$;
8:             **else**
9:                 $\{\phi^{(l)}; \psi_p^r\} \leftarrow out\_queue(\mathcal{M}^{r-1})$;
10:                **if** $r\%B == 0$ **then**
11:                    $\{\phi^{(l)}; \psi_p^r\} \leftarrow \{\phi^{r-1}, \psi^{r-1}\}$;
12:                **end if**
13:                $\psi_p^r, \phi_p^r \leftarrow$ **Client**$(p, r, \psi_p^r, \phi^{(l)}, \phi^{r-1})$;
14:                $\omega_p^r \leftarrow \omega^{r-1}$;
15:            **end if**
16:        **end for**
17:        Press $\cup_{p \in \mathcal{C}} \{\phi_p^r; \psi_p^r\}$ into queue $\mathcal{M}^r$;
18:        $\{\psi^r; \omega^r; \phi^r\} \leftarrow \sum_{p \in \mathcal{C}} \frac{|\mathcal{D}_p|}{\sum_{p \in \mathcal{C}} |\mathcal{D}_p|} \{\psi_p^r; \omega_p^r; \phi_p^r\}$;
19:    **end for**
20:    **return** The global model $\Theta = \{\Psi, \Omega, \Phi\}$.

---

**Algorithm 2** The Client Updates for FedMP

---

**Input**: The index of client $p$, the current round number $r$, the downloaded models $S_1, S_2, S_3$ and the hyperparameters of Algorithm 1.

**Output**: The local updated model $\Theta_p$.

1: **Client**$(p, r, S_1, S_2, S_3)$:
2:     **if** $r \leq R_{it}$ **then**
3:         $\psi_p^r, \omega_p^r, \phi_p^r \leftarrow S_1, S_2, S_3$;
4:     **else**
5:         $\psi_p^r, \phi_p^{(l),r}, \phi_p^{(g),r} \leftarrow S_1, S_2, S_3$;
6:     **end if**
7:     **for** $e = 1 \rightarrow E$ **do**
8:         **for** each batch $\mathcal{B} \in \mathcal{D}_p$ **do**
9:             **if** $r \leq R_{it}$ **then**
10:                $\mathcal{L} \leftarrow$ Eq. (9) with $\mathcal{B}$;
11:                $\Theta_p : \{\psi_p^r; \omega_p^r; \phi_p^r\} \leftarrow \{\psi_p^r; \omega_p^r; \phi_p^r\} - \eta \nabla \mathcal{L}$;
12:            **else**
13:                $\mathcal{L} \leftarrow$ Eq. (8) with $\mathcal{B}$;
14:                $\Theta_p : \{\psi_p^r; \phi_p^{(l),r}; \phi_p^{(g),r}\} \leftarrow \{\psi_p^r; \phi_p^{(l),r}; \phi_p^{(g),r}\} - \eta \nabla \mathcal{L}$;
15:            **end if**
16:        **end for**
17:    **end for**
18:    **return** The local updated model $\Theta_p$.

---

# D   Proof of Theorem 1

In this section, we provide a convergence analysis of the proposed FedMP. First, some assumptions are introduced to help complete the following theoretical analysis.

**Assumption 1.** *For any $p \in [P]$, local loss function for local solution path $\mathcal{L}_p^{(l)}$ and local loss function for global solution path $\mathcal{L}_p^{(g)}$ are L-smooth with respect to $\Theta$. For $\forall \, \Theta$ and $\Theta'$, the following inequalities hold:*

$$\|\nabla \mathcal{L}_p^{(l)}(\Theta) - \nabla \mathcal{L}_p^{(l)}(\Theta')\| \le L_1 \|\Theta - \Theta'\|, \tag{17}$$

$$\|\nabla \mathcal{L}_p^{(g)}(\Theta) - \nabla \mathcal{L}_p^{(g)}(\Theta')\| \le L_2 \|\Theta - \Theta'\|, \tag{18}$$

*where $L_1$ and $L_2$ are Lipschitz constants.*

**Assumption 2.** *The upper bound on the variances of the local gradient to the aggregated mean can be given as follows*

$$\frac{1}{P} \sum_{p=1}^{P} \|\nabla \mathcal{L}_p^{(l)}(\Theta) - \nabla \mathcal{L}^{(l)}(\Theta)\|^2 \le \delta_L^2, \tag{19}$$

$$\frac{1}{P} \sum_{p=1}^{P} \|\nabla \mathcal{L}_p^{(g)}(\Theta) - \nabla \mathcal{L}^{(g)}(\Theta)\|^2 \le \delta_G^2, \tag{20}$$

*where $\delta_L$ and $\delta_G$ are the constants.*

Since the loss functions depend on the visible data, and the use of the swapping strategy results in the equivalent of extending the gradient from the sampling data, we here set the step size of $r$ to the $B$ communication rounds. Since only random clients are activated in each communication round, we need to introduce the following lemmas before deriving the convergence analysis.

**Lemma 1.** *If Assumption 2 holds, the upper bound on the variances of the local gradient from the random clients to the aggregated mean can be given as follows*

$$\mathbb{E}_{\mathcal{C}^r} \left[ \|\frac{1}{K} \sum_{p \in \mathcal{C}^r} \nabla \mathcal{L}_p(\Theta^r) - \nabla \mathcal{L}(\Theta^r)\|^2 \right] \le \frac{2(P-K)}{K(P-1)} \left( \delta_L^2 + \gamma^2 \delta_G^2 \right), \tag{21}$$

*where $\mathcal{C}^r$ is the selected clients set of the $r$-th round of communication.*

*Proof.* First, we give the following equation

$$\mathbb{E}_{\mathcal{C}^r} \left[ \|\frac{1}{K} \sum_{p \in \mathcal{C}^r} \nabla \mathcal{L}_p(\Theta^r) - \nabla \mathcal{L}(\Theta^r)\|^2 \right]$$

$$= \mathbb{E}_{\mathcal{C}^r} \left[ \|\frac{1}{K} \sum_{p \in \mathcal{C}^r} \left( \nabla \mathcal{L}_p(\Theta^r) - \nabla \mathcal{L}(\Theta^r) \right)\|^2 \right]$$

$$= \frac{1}{K^2} \mathbb{E}_{\mathcal{C}^r} \left[ \|\sum_{p \in \mathcal{C}^r} \left( \nabla \mathcal{L}_p(\Theta^r) - \nabla \mathcal{L}(\Theta^r) \right)\|^2 \right] \tag{22}$$

$$= \frac{1}{K^2} \mathbb{E}_{\mathcal{C}^r} \left[ \sum_{p \in \mathcal{C}^r} \|\nabla \mathcal{L}_p(\Theta^r) - \nabla \mathcal{L}(\Theta^r)\|^2 \right.$$

$$\left. + \sum_{p \in \mathcal{C}^r} \sum_{p' \in \mathcal{C}^r, p \ne p'} \langle \nabla \mathcal{L}_p(\Theta^r) - \nabla \mathcal{L}(\Theta^r), \nabla \mathcal{L}_{p'}(\Theta^r) - \nabla \mathcal{L}(\Theta^r) \rangle \right].$$

Considering that in each communication round, each client is selected with equal probability, the selection probability terms for the two terms in Equation 22 are $\frac{K}{P}$ and $\frac{K(K-1)}{P(P-1)}$. Moreover, due to

$\mathcal{L}(\Theta) = \frac{1}{P}\sum_{p=1}^{P}\mathcal{L}_p(\Theta)$, the following equation holds:

$$\|\frac{1}{P}\sum_{p=1}^{P}\nabla\mathcal{L}_p(\Theta) - \nabla\mathcal{L}(\Theta)\|^2$$

$$=\frac{1}{P}\sum_{p=1}^{P}\|\nabla\mathcal{L}_p(\Theta) - \nabla\mathcal{L}(\Theta)\|^2 \tag{23}$$

$$+\frac{1}{P}\sum_{p=1}^{P}\sum_{p\neq p'}\langle\nabla\mathcal{L}_p(\Theta) - \nabla\mathcal{L}(\Theta), \nabla\mathcal{L}_{p'}(\Theta) - \nabla\mathcal{L}(\Theta)\rangle$$

$$=0.$$

So that,

$$\sum_{p=1}^{P}\|\nabla\mathcal{L}_p(\Theta) - \nabla\mathcal{L}(\Theta)\|^2 = -\sum_{p=1}^{P}\sum_{p\neq p'}\langle\nabla\mathcal{L}_p(\Theta) - \nabla\mathcal{L}(\Theta), \nabla\mathcal{L}_{p'}(\Theta) - \nabla\mathcal{L}(\Theta)\rangle. \tag{24}$$

Therefore, the following formula can be derived

$$\mathbb{E}_{\mathcal{C}^r}\left[\|\frac{1}{K}\sum_{p\in\mathcal{C}^r}\nabla\mathcal{L}_p(\Theta^r) - \nabla\mathcal{L}(\Theta^r)\|^2\right]$$

$$=\frac{1}{K^2}\left[\frac{K}{P}\sum_{p=1}^{P}\|\nabla\mathcal{L}_p(\Theta^r) - \nabla\mathcal{L}(\Theta^r)\|^2\right.$$

$$\left.+\frac{K(K-1)}{P(P-1)}\sum_{p=1}^{P}\sum_{p\neq p'}\langle\nabla\mathcal{L}_p(\Theta^r) - \nabla\mathcal{L}(\Theta^r), \nabla\mathcal{L}_{p'}(\Theta^r) - \nabla\mathcal{L}(\Theta^r)\rangle\right]$$

$$=\frac{P-K}{KP(P-1)}\sum_{p=1}^{P}\|\nabla\mathcal{L}_p(\Theta^r) - \nabla\mathcal{L}(\Theta^r)\|^2 \tag{25}$$

$$=\frac{P-K}{K(P-1)}\frac{1}{P}\sum_{p=1}^{P}\|\nabla\mathcal{L}_p^{(l)}(\Theta^r) + \gamma\nabla\mathcal{L}_p^{(g)}(\Theta^r) - \nabla\mathcal{L}^{(l)}(\Theta^r) - \gamma\nabla\mathcal{L}^{(g)}(\Theta^r)\|^2$$

$$\leq\frac{P-K}{K(P-1)}\frac{1}{P}\sum_{p=1}^{P}\left[2\|\nabla\mathcal{L}_p^{(l)}(\Theta^r) - \nabla\mathcal{L}^{(l)}(\Theta^r)\|^2 + 2\gamma^2\|\nabla\mathcal{L}_p^{(g)}(\Theta^r) - \nabla\mathcal{L}^{(g)}(\Theta^r)\|^2\right]$$

$$\leq\frac{2(P-K)}{K(P-1)}\left(\delta_L^2 + \gamma^2\delta_G^2\right).$$

$\square$

**Lemma 2.** *If learning rate* $\eta \leq \frac{1}{2EB\sqrt{L_1^2+\gamma^2 L_2^2}}$ *and Assumption 1 holds, it can be derived that*

$$\frac{1}{EB}\sum_{e=0}^{EB-1}\|\Theta_{p,e}^r - \Theta^r\|^2 \leq 8E^2B^2\eta^2\|\mathcal{L}_p(\Theta^r)\|^2, \tag{26}$$

*where* $\Theta_{p,e}^r$ *means the e-th stage in r-th communication round at p-th client, and* $\Theta_{p,0}^r = \Theta^r$

*Proof.* To begin with, we need to introduce the following inequality

$$(a+b)^2 \leq (1+\frac{1}{c})a^2 + (1+c)b^2, \tag{27}$$

where $c > 0$ is a positive number. Equation 27 can be easily proved by

$$
\begin{aligned}
a^2 + 2ab + b^2 \leq &a^2 + \frac{1}{c}a^2 + b^2 + cb^2, \\
&\Downarrow \\
0 \leq &\frac{1}{c}a^2 + cb^2 - 2ab, \\
&\Downarrow \\
0 \leq &(\sqrt{\frac{1}{c}}a - \sqrt{c}b)^2.
\end{aligned}
\tag{28}
$$

In each update stage, we can represent the process of updating as $\Theta_{p,e}^r = \Theta_{p,e-1}^r - \eta \nabla \mathcal{L}(\Theta_{p,e-1}^r)$. Therefore, based on the above, we can derive the following equation

$$
\begin{aligned}
&\|\Theta_{p,e}^r - \Theta^r\|^2 \\
=&\|\Theta_{p,e-1}^r - \eta \nabla \mathcal{L}(\Theta_{p,e-1}^r) - \Theta^r\|^2 \\
=&\|\Theta_{p,e-1}^r - \eta \nabla \mathcal{L}(\Theta_{p,e-1}^r) + \eta \nabla \mathcal{L}_p(\Theta^r) - \eta \nabla \mathcal{L}_p(\Theta^r) - \Theta^r\|^2 \\
\leq&(1 + \frac{1}{EB})\|\Theta_{p,e-1}^r - \eta \nabla \mathcal{L}_p(\Theta^r) - \Theta^r\|^2 \\
&+ (1 + EB)\|\eta \nabla \mathcal{L}_p(\Theta^r) - \eta \nabla \mathcal{L}(\Theta_{p,e-1}^r)\|^2 \\
\leq&(1 + \frac{1}{EB})\left[(1 + \frac{1}{2EB})\|\Theta_{p,e-1}^r - \Theta^r\|^2 + (1 + 2EB)\|\eta \nabla \mathcal{L}_p(\Theta^r)\|^2\right] \\
&+ (1 + EB)\left[2\eta^2\|\nabla \mathcal{L}_p^{(l)}(\Theta^r) - \nabla \mathcal{L}_p^{(l)}(\Theta_{p,e-1}^r)\|^2\right. \\
&\left.+ 2\eta^2\gamma^2\|\nabla \mathcal{L}_p^{(g)}(\Theta^r) - \nabla \mathcal{L}_p^{(g)}(\Theta_{p,e-1}^r)\|^2\right] \\
\leq&(1 + \frac{1}{EB})(1 + \frac{1}{2EB} + 2EB\eta^2 L_1^2 + 2EB\eta^2\gamma^2 L_2^2)\|\Theta_{p,e-1}^r - \Theta^r\|^2 \\
&+ (1 + \frac{1}{EB})(1 + \frac{1}{2EB})\eta^2\|\nabla \mathcal{L}_p(\Theta^r)\|^2.
\end{aligned}
\tag{29}
$$

When $\eta \leq \frac{1}{2EB\sqrt{L_1^2 + \gamma^2 L_2^2}}$, the above equation can be written as

$$
\begin{aligned}
&\|\Theta_{p,e}^r - \Theta^r\|^2 \\
\leq&(1 + \frac{1}{EB})^2\|\Theta_{p,e-1}^r - \Theta^r\|^2 + (1 + \frac{1}{EB})(1 + \frac{1}{2EB})\eta^2\|\nabla \mathcal{L}_p(\Theta^r)\|^2 \\
\leq&\sum_{i=0}^{e-1}(1 + \frac{1}{EB})^{2i+1}(1 + 2EB)\eta^2\|\nabla \mathcal{L}_p(\Theta^r)\|^2 \\
=&(1 + 2EB)\eta^2\|\nabla \mathcal{L}_p(\Theta^r)\|^2 \frac{(1 + \frac{1}{EB})^{2e+1} - (1 + \frac{1}{EB})}{(1 + \frac{1}{EB})^2 - 1} \\
\leq&(1 + 2EB)\eta^2\|\nabla \mathcal{L}_p(\Theta^r)\|^2 \frac{(1 + \frac{1}{EB})^{2e+1}}{\frac{2}{EB} + \frac{1}{E^2 B^2}} \\
=&E^2 B^2 (1 + \frac{1}{EB})^{2e+1}\eta^2\|\nabla \mathcal{L}_p(\Theta^r)\|^2 \\
=&EB(1 + EB)(1 + \frac{1}{EB})^{2e}\eta^2\|\nabla \mathcal{L}_p(\Theta^r)\|^2.
\end{aligned}
\tag{30}
$$

Therefore, we can derive that

$$
\begin{aligned}
\frac{1}{EB}\sum_{e=0}^{EB-1}\|\Theta_{p,e}^r - \Theta^r\|^2 &\leq EB(1+EB)\eta^2\|\nabla\mathcal{L}_p(\Theta^r)\|^2\frac{1}{EB}\sum_{e=0}^{EB-1}(1+\frac{1}{EB})^{2e}\\
&\leq (1+EB)\eta^2\|\nabla\mathcal{L}_p(\Theta^r)\|^2\frac{(1+\frac{1}{EB})^{2e}-1}{\frac{2}{EB}+\frac{1}{E^2B^2}}\\
&\leq \frac{E^2B^2(1+EB)}{2EB+1}\eta^2\|\nabla\mathcal{L}_p(\Theta^r)\|^2(1+\frac{1}{EB})^{2EB}\\
&\leq \frac{EB(1+EB)}{2}\eta^2\|\nabla\mathcal{L}_p(\Theta^r)\|^2(1+\frac{1}{EB})^{2EB}\\
&\leq E^2B^2\eta^2\|\nabla\mathcal{L}_p(\Theta^r)\|^2\lim_{EB\to\infty}(1+\frac{1}{EB})^{2EB}\\
&\leq 8E^2B^2\eta^2\|\nabla\mathcal{L}_p(\Theta^r)\|^2.
\end{aligned}
\tag{31}
$$

$\square$

**Theorem 1.** *Suppose Assumption 1 and 2 hold, the convergence property of the proposed method can be described by*

$$
\frac{1}{R}\sum_{r=0}^{R-1}\mathbb{E}\left[\|\nabla\mathcal{L}(\Theta^r)\|^2\right]\leq\epsilon,
\tag{32}
$$

*where $\epsilon = \frac{\mathbb{E}[\mathcal{L}(\Theta^0)-\mathcal{L}(\Theta^R)]}{\mathcal{H}R}+\mathcal{S}$, and*

$$
\begin{aligned}
\mathcal{H} = \frac{\eta EB}{2}&\left[1-2\eta EB(L_1+\gamma L_2)-32\eta^2 E^2B^2(L_1+\gamma L_2)^2\right.\\
&\left.-64\eta^3 E^3B^3(L_1+\gamma L_2)^3\right],
\end{aligned}
\tag{33}
$$

$$
\begin{aligned}
\mathcal{S} = \left[\frac{\eta EB}{2}+\eta^2 E^2B^2(L_1+\gamma L_2)\right]&\left[64\eta^2 E^2B^2(L_1+\gamma L_2)^2+\frac{4}{K}\right]\\
&\frac{P-K}{P-1}(\delta_L^2+\gamma^2\delta_G^2).
\end{aligned}
\tag{34}
$$

From Theorem Theorem 1 we can know that the proposed FedMP can reach the convergence with appropriate hyperparameters choices.

*Proof.* First of all, based on the above lemmas and assumptions, we can derive the following equations

$$
\begin{aligned}
\Theta^{r+1} &= \frac{1}{K}\sum_{p\in\mathcal{C}^r}\Theta_{p,EB}^r\\
&= \frac{1}{K}\sum_{p\in\mathcal{C}^r}\left[\Theta^r-\eta\sum_{e=0}^{EB-1}\nabla\mathcal{L}_p(\Theta_{p,e}^r)\right]\\
&= \Theta^r-\frac{\eta EB}{EBK}\sum_{p\in\mathcal{C}^r}\sum_{e=0}^{EB-1}\nabla\mathcal{L}_p(\Theta_{p,e}^r) = \Theta^r-\eta EB\mathcal{G}^r,
\end{aligned}
\tag{35}
$$

and

$$
\begin{aligned}
&\|\nabla\mathcal{L}(\Theta)-\nabla\mathcal{L}(\Theta')\|\\
=&\|\frac{1}{P}\sum_{p=1}^P\nabla\mathcal{L}_p(\Theta)-\frac{1}{P}\sum_{p=1}^P\nabla\mathcal{L}_p(\Theta')\|\\
=&\frac{1}{P}\sum_{p=1}^P\|\nabla\mathcal{L}_p^{(l)}(\Theta)+\gamma\nabla\mathcal{L}_p^{(g)}(\Theta)-\nabla\mathcal{L}_p^{(l)}(\Theta')-\gamma\nabla\mathcal{L}_p^{(g)}(\Theta')\|\\
\leq&(L_1+\gamma L_2)\|\Theta-\Theta'\|.
\end{aligned}
\tag{36}
$$

Furthermore, Equation 36 can also be written as the second-order Taylor expansion form

$$\mathcal{L}(\Theta) \leq \mathcal{L}(\Theta') + \langle \Theta - \Theta', \nabla\mathcal{L}(\Theta') \rangle + \frac{L_1 + \gamma L_2}{2} \|\Theta - \Theta'\|^2. \tag{37}$$

Hence, the upper bound of global optimization objective for each round can be derived as

$$
\begin{aligned}
&\mathbb{E}_{\mathcal{C}^r}\left[\mathcal{L}(\Theta^{r+1}) - \mathcal{L}(\Theta^r)\right]\\
&\leq \mathbb{E}_{\mathcal{C}^r}\left[\langle \Theta^{r+1} - \Theta^r, \nabla\mathcal{L}(\Theta^r)\rangle\right] + \frac{L_1 + \gamma L_2}{2}\mathbb{E}_{\mathcal{C}^r}\left[\|\Theta^{r+1} - \Theta^r\|^2\right]\\
&= \eta E B \mathbb{E}_{\mathcal{C}^r}\left[\langle -\mathcal{G}^r, \nabla\mathcal{L}(\Theta^r)\rangle\right] + \frac{L_1 + \gamma L_2}{2}\mathbb{E}_{\mathcal{C}^r}\left[\|\mathcal{G}^r\|^2\right]\eta^2 E^2 B^2\\
&= \eta E B \mathbb{E}_{\mathcal{C}^r}\left[\langle \nabla\mathcal{L}(\Theta^r) - \mathcal{G}^r - \nabla\mathcal{L}(\Theta^r), \nabla\mathcal{L}(\Theta^r)\rangle\right]\\
&\quad + \frac{\eta^2 E^2 B^2 (L_1 + \gamma L_2)}{2}\mathbb{E}_{\mathcal{C}^r}\left[\|\mathcal{G}^r\|^2\right]\\
&= -\eta E B \mathbb{E}_{\mathcal{C}^r}\left[\|\nabla\mathcal{L}(\Theta^r)\|^2\right] - \eta E B \mathbb{E}_{\mathcal{C}^r}\left[\langle \mathcal{G}^r - \nabla\mathcal{L}(\Theta^r), \nabla\mathcal{L}(\Theta^r)\rangle\right]\\
&\quad + \frac{\eta^2 E^2 B^2 (L_1 + \gamma L_2)}{2}\mathbb{E}_{\mathcal{C}^r}\left[\|\mathcal{G}^r\|^2\right]\\
&\leq -\frac{\eta E B}{2}\mathbb{E}_{\mathcal{C}^r}\left[\|\nabla\mathcal{L}(\Theta^r)\|^2\right] + \frac{\eta E B}{2}\mathbb{E}_{\mathcal{C}^r}\left[\|\mathcal{G}^r - \nabla\mathcal{L}(\Theta^r)\|^2\right]\\
&\quad + \frac{\eta^2 E^2 B^2 (L_1 + \gamma L_2)}{2}\mathbb{E}_{\mathcal{C}^r}\left[\|\mathcal{G}^r - \nabla\mathcal{L}(\Theta^r) + \nabla\mathcal{L}(\Theta^r)\|^2\right]\\
&\leq -\frac{\eta E B}{2}\left[1 - 2\eta E B(L_1 + \gamma L_2)\right]\mathbb{E}_{\mathcal{C}^r}\left[\|\nabla\mathcal{L}(\Theta^r)\|^2\right]\\
&\quad + \left[\frac{\eta E B}{2} + \eta^2 E^2 B^2 (L_1 + \gamma L_2)\right]\mathbb{E}_{\mathcal{C}^r}\left[\|\mathcal{G}^r - \nabla\mathcal{L}(\Theta^r)\|^2\right],
\end{aligned}
\tag{38}
$$

here, $\mathbb{E}_{\mathcal{C}^r}\left[\|\mathcal{G}^r - \nabla\mathcal{L}(\Theta^r)\|^2\right]$ can be written as

$$
\begin{aligned}
&\mathbb{E}_{\mathcal{C}^r}\left[\|\mathcal{G}^r - \nabla\mathcal{L}(\Theta^r)\|^2\right]\\
&= \mathbb{E}_{\mathcal{C}^r}\left[\|\frac{1}{EBK}\sum_{p\in\mathcal{C}^r}\sum_{e=0}^{EB-1}\nabla\mathcal{L}_p(\Theta_{p,e}^r) - \frac{1}{EBK}\sum_{p\in\mathcal{C}^r}\sum_{e=0}^{EB-1}\nabla\mathcal{L}_p(\Theta^r)\right.\\
&\quad \left. + \frac{1}{EBK}\sum_{p\in\mathcal{C}^r}\sum_{e=0}^{EB-1}\nabla\mathcal{L}_p(\Theta^r) - \nabla\mathcal{L}(\Theta^r)\|^2\right]\\
&\leq 2\mathbb{E}_{\mathcal{C}^r}\left[\|\frac{1}{EBK}\sum_{p\in\mathcal{C}^r}\sum_{e=0}^{EB-1}(\nabla\mathcal{L}_p(\Theta_{p,e}^r) - \nabla\mathcal{L}_p(\Theta^r))\|^2\right]\\
&\quad + 2\mathbb{E}_{\mathcal{C}^r}\left[\|\frac{1}{K}\sum_{p\in\mathcal{C}^r}\nabla\mathcal{L}_p(\Theta^r) - \nabla\mathcal{L}(\Theta^r)\|^2\right]\\
&\leq 2\mathbb{E}_{\mathcal{C}^r}\left[\frac{1}{K}\sum_{p\in\mathcal{C}^r}\frac{1}{EB}\sum_{e=0}^{EB-1}(L_1 + \gamma L_2)^2\|\Theta_{p,e}^r - \Theta^r\|^2\right]\\
&\quad + 2\mathbb{E}_{\mathcal{C}^r}\left[\|\frac{1}{K}\sum_{p\in\mathcal{C}^r}\nabla\mathcal{L}_p(\Theta^r) - \nabla\mathcal{L}(\Theta^r)\|^2\right]\\
&\leq 16 E^2 B^2 \eta^2 (L_1 + \gamma L_2)^2 \mathbb{E}_{\mathcal{C}^r}\left[\frac{1}{K}\sum_{p\in\mathcal{C}^r}\|\nabla\mathcal{L}_p(\Theta^r)\|^2\right] + \frac{4(P-K)}{K(P-1)}\left(\delta_L^2 + \gamma^2\delta_G^2\right)
\end{aligned}
\tag{39}
$$

$$\leq 16E^2B^2\eta^2(L_1+\gamma L_2)^2\mathbb{E}_{\mathcal{C}^r}\left[\frac{1}{K}\sum_{p\in\mathcal{C}^r}\|\nabla\mathcal{L}_p(\Theta^r)-\nabla\mathcal{L}(\Theta^r)+\nabla\mathcal{L}(\Theta^r)\|^2\right]$$

$$+\frac{4(P-K)}{K(P-1)}\left(\delta_L^2+\gamma^2\delta_G^2\right)$$

$$\leq 32E^2B^2\eta^2(L_1+\gamma L_2)^2\mathbb{E}_{\mathcal{C}^r}\left[\|\nabla\mathcal{L}(\Theta^r)\|^2\right]$$

$$+\left[64E^2B^2\eta^2(L_1+\gamma L_2)^2+\frac{4}{K}\right]\frac{(P-K)}{(P-1)}\left(\delta_L^2+\gamma^2\delta_G^2\right).$$

Therefore, Equation 38 can be written as

$$\mathbb{E}_{\mathcal{C}^r}\left[\mathcal{L}(\Theta^{r+1})-\mathcal{L}(\Theta^r)\right]$$
$$\leq-\frac{\eta EB}{2}\left[1-2\eta EB(L_1+\gamma L_2)-32E^2B^2\eta^2(L_1+\gamma L_2)^2\right.$$
$$\left.-64E^3B^3\eta^3(L_1+\gamma L_2)^3\right]\mathbb{E}_{\mathcal{C}^r}\left[\|\nabla\mathcal{L}(\Theta^r)\|^2\right] \tag{40}$$
$$+\left[\frac{\eta EB}{2}+\eta^2E^2B^2(L_1+\gamma L_2)\right]\left[64E^2B^2\eta^2(L_1+\gamma L_2)^2+\frac{4}{K}\right]$$
$$\frac{(P-K)}{(P-1)}\left(\delta_L^2+\gamma^2\delta_G^2\right).$$

We let

$$\mathcal{H}=\frac{\eta EB}{2}\left[1-2\eta EB(L_1+\gamma L_2)-32\eta^2E^2B^2(L_1+\gamma L_2)^2\right.$$
$$\left.-64\eta^3E^3B^3(L_1+\gamma L_2)^3\right], \tag{41}$$

and

$$\mathcal{S}=\left[\frac{\eta EB}{2}+\eta^2E^2B^2(L_1+\gamma L_2)\right]\left[64\eta^2E^2B^2(L_1+\gamma L_2)^2+\frac{4}{K}\right]$$
$$\frac{P-K}{P-1}(\delta_L^2+\gamma^2\delta_G^2). \tag{42}$$

So Equation 40 can be written as

$$\mathbb{E}_{\mathcal{C}^r}\left[\mathcal{L}(\Theta^{r+1})-\mathcal{L}(\Theta^r)\right]\leq-\mathcal{H}\mathbb{E}_{\mathcal{C}^r}\left[\|\nabla\mathcal{L}(\Theta^r)\|^2\right]+\mathcal{S}. \tag{43}$$

When summing over all communication rounds, Equation 43 can be expanded as

$$\frac{1}{R}\sum_{r=0}^{R-1}\mathbb{E}\left[\|\nabla\mathcal{L}(\Theta^r)\|^2\right]\leq\frac{\mathbb{E}[\mathcal{L}(\Theta^0)-\mathcal{L}(\Theta^R)]}{\mathcal{H}R}+\mathcal{S}. \tag{44}$$

□

# E Supplementary Experiments

## E.1 Training Setup

The distributions of the train set labels on each client under different levels of data heterogeneity are visualized in Figure 8. The partitioned train sets are placed on the clients and the test sets are placed on the server for testing the global model.

**Other settings:** ResNet-18 [51] is used for the experiments. We implemented the proposed method using PyTorch 1.12, and deployed it on a machine configured with AMD R9 5900X, 64GB memory, and Nvidia RTX3090. For the CIFAR-10 and CIFAR-100 datasets, we train in a system with 20 clients and the batch sizes of the data are set to 128 and 64, respectively. For the Flowers102 dataset, we use a system with 10 clients and the batch size of the data is set to 64. 50% of the clients are chosen in each round for training and aggregation, and the number of local epochs $E$ is set to 5.

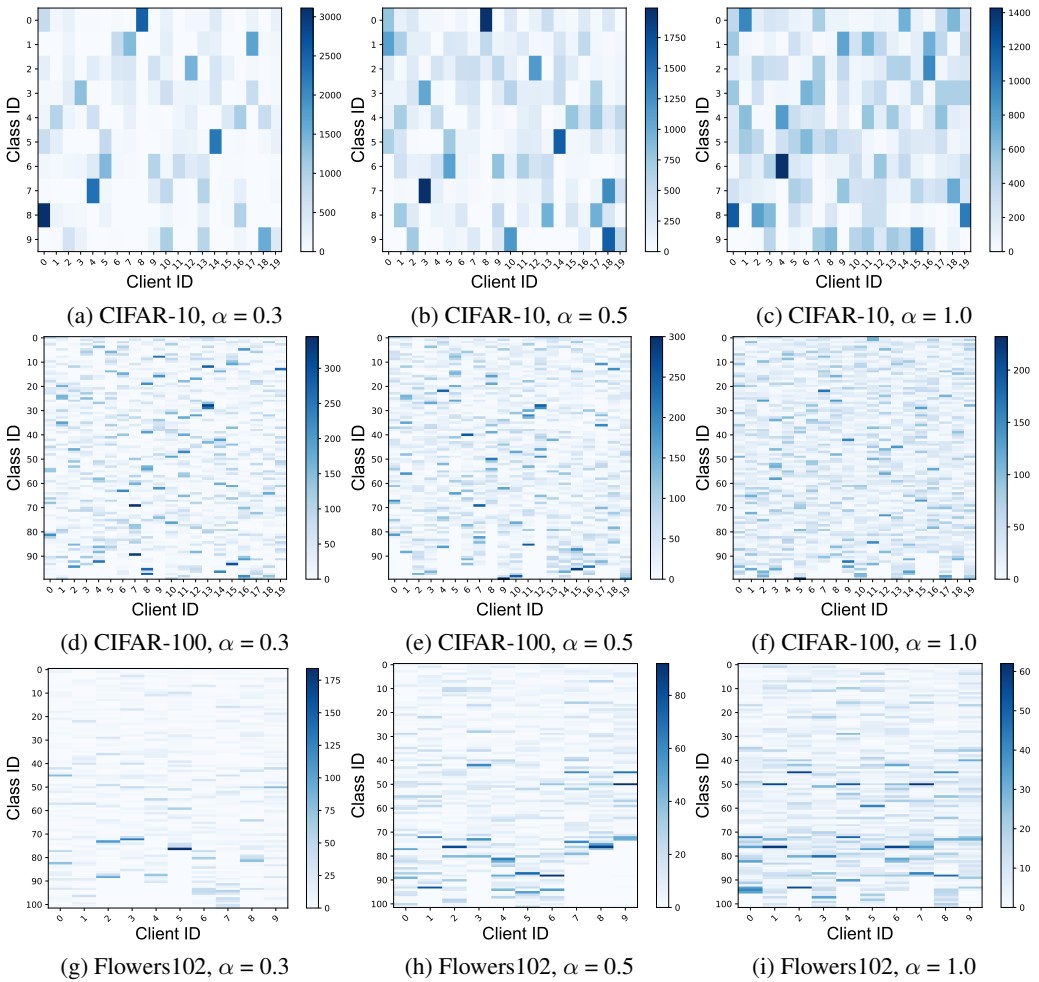

Figure 8: An illustration for the distributions of labels in the train set at each client. The above figures are obtained under three heterogeneous levels on the CIFAR-10, CIFAR-100 and Flowers102 datasets, respectively.

We set the balance round $B$ to 5, for the CIFAR-10 dataset we set the initial training round $R_{it}$ to 5, and for the CIFAR-100 and Flowers102 datasets we set the initial training round $R_{it}$ to 10. To speed up the initialization progress, we constrain the Lipschitz smoothness for the initial model. The hyperparameter $\gamma$ is set to 0.1, and SGD optimizer with learning rate $\eta = 0.05$ is used. The main components of the network are listed in Table 5, where ConvT means the transposed convolution operator, i.e. Deconvolutional Networks [52]. Under the above settings, we run the proposed method to generate the final results.

### E.2 More Results

In order to verify that whether the proposed method works properly, we output the test accuracies of the global model produced by FedMP during the training process and show it in Figure 9 with the results of the comparative methods. It can be seen that the proposed method can make the global model output a more stable test accuracy after a period of training. Moreover, the proposed method can achieve the higher classification accuracies than the other comparative methods.

The results for more clients are displayed in Figure 10. Similar to the results of Client 1, the models obtained via the multiple paths solving approach are able to obtain better accuracy on the data at other clients, which demonstrates that the multiple paths solving approach is effective in preserving global information and reaching a soft consensus between the global and the local.

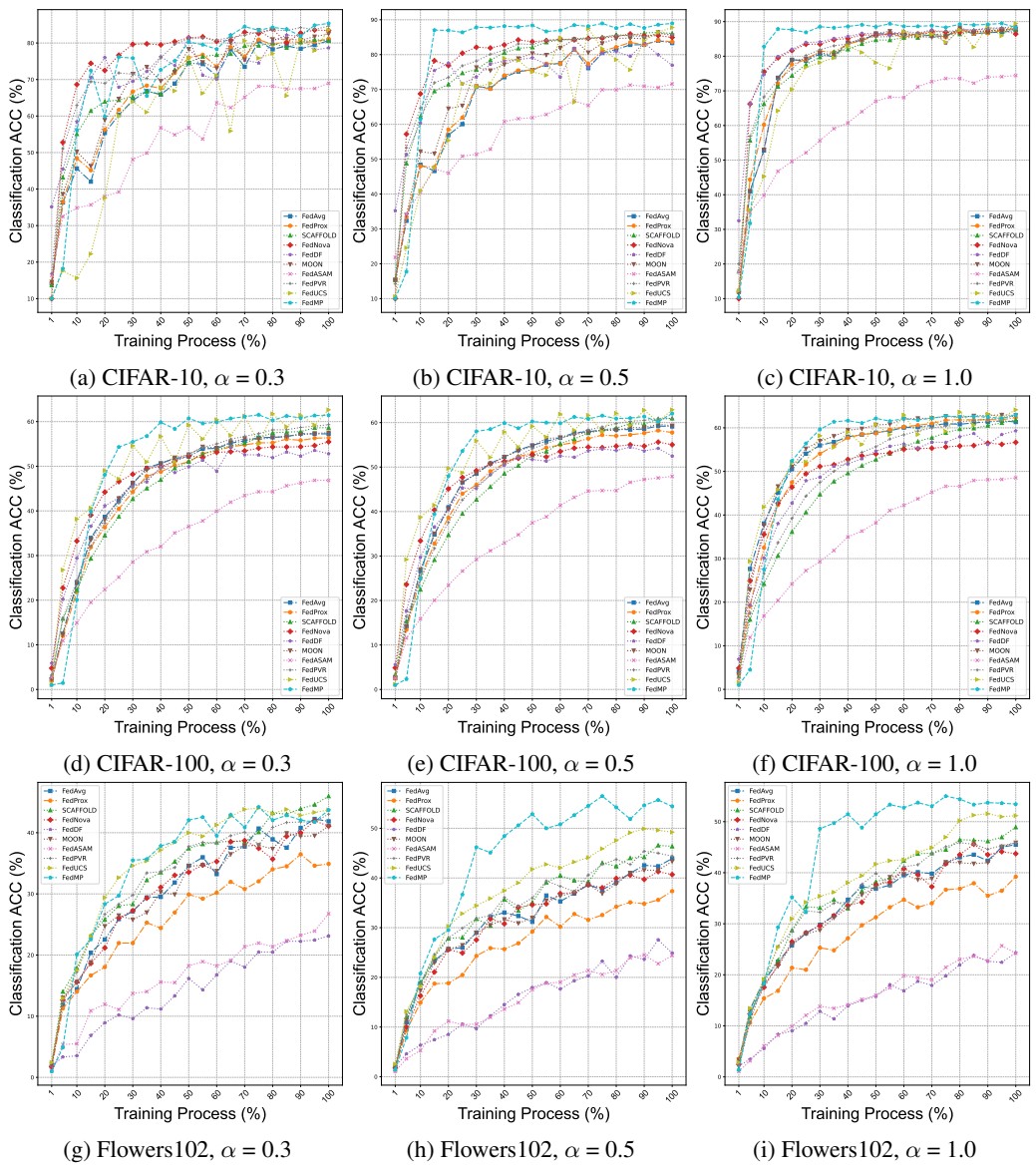

Figure 9: Classification accuracy curves of the global model on the test set with the training process. It can be seen that the proposed method is able to converge on all dataset settings.

Table 5: Network architecture for three test datasets. The gray background represents the parameters that need to be communicated.

(a) Network architecture for CIFAR-10.

| Layer / Kernel / Stride | | Output shape | # Params |
|---|---|---|---|
| Path-local | Path-global | - | - |
| ConvT / 3×3 / 3 | ConvT / 3×3 / 3 | 64×96×96 | 1.7k×2 |
| ConvT / 34×34 / 2 | ConvT / 34×34 / 2 | 3×224×224 | 0.2M×2 |
| ResNet-18 / - / - | | 1,000 | 11.7M |
| FC / - / - | | 100 | 0.1M |
| FC / - / - | | 10 | 1k |

(b) Network architecture for CIFAR-100.

| Layer / Kernel / Stride | | Output shape | # Params |
|---|---|---|---|
| Path-local | Path-global | - | - |
| ConvT / 3×3 / 3 | ConvT / 3×3 / 3 | 64×96×96 | 1.7k×2 |
| ConvT / 34×34 / 2 | ConvT / 34×34 / 2 | 3×224×224 | 0.2M×2 |
| ResNet-18 / - / - | | 1,000 | 11.7M |
| FC / - / - | | 100 | 0.1M |
| FC / - / - | | 100 | 10k |

(c) Network architecture for Flowers102.

| Layer / Kernel / Stride | | Output shape | # Params |
|---|---|---|---|
| Path-local | Path-global | - | - |
| Conv / 3×3 / 1 | Conv / 3×3 / 1 | 64×222×222 | 1.7k×2 |
| ConvT / 3×3 / 1 | ConvT / 3×3 / 1 | 3×224×224 | 1.7k×2 |
| ResNet-18 / - / - | | 1,000 | 11.7M |
| FC / - / - | | 100 | 0.1M |
| FC / - / - | | 102 | 10.2k |

We show the detailed results on CIFAR-100 in Figure 11. The curves represent the performance of the models achieved on the local dataset after each communication round of local training and global aggregation in turn during the training process. The fluctuation of the curves represents the characteristic that the model oscillates between global and local features during the federated training process. After local training, the local features dominate and thus, the performance on the local dataset is improved. Whereas, after aggregation, global features gain dominance and hence, the performance on the local dataset decreases. The curve of using the multiple paths solving approach has less fluctuation because the model obtained with it seeks consensus among global and local features at the local training stage, therefore, although it cannot fit the current dataset better, after aggregation, the model obtained will not lose too much local knowledge because the local training on other clients also takes into account the reaching of consensus. Models trained by multiple paths solving methods have smaller training fluctuations, which means that there is less feature struggle during training and the federated process will be able to perform more efficiently. Using the correct multiple paths can help the local model to reduce the vanishing of local features and keep more global features.

### E.3 FedMP with Full Multiple Paths

Here, instead of using the communication cost reduction method proposed in Section 2.4 of the main body, we use a multiple paths solving method for the complete task network. We use the complete task network here as the mapping with global feature preferences and the mapping with local feature preferences, respectively, and use the same shared projection as in the experiments in the main text to motivate a soft consensus. The network structure is shown in Table 6. We show the classification accuracy results in Table 7 along with the results in the main text. It can be seen that the results obtained are further improved after using the full multiple paths (FedMP-F). However, the penalty for this enhancement would be a higher communication cost, and we believe that the small amount of performance degradation associated with the reduction in communication cost is acceptable. Therefore, there is flexibility to trade-off between the two approaches depending on the requirements.

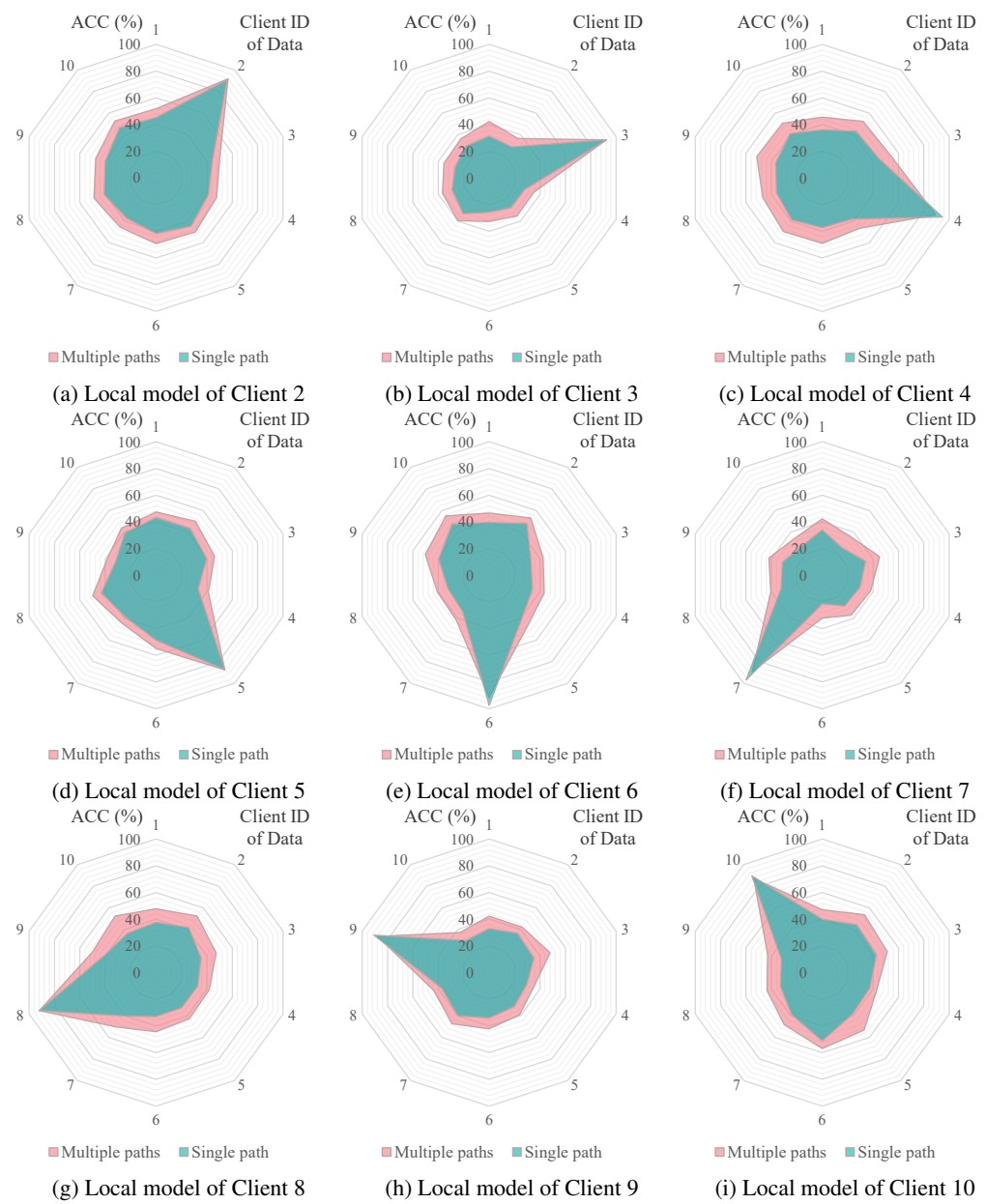

Figure 10: Classification accuracies of each local models on the data in different clients. The local model trained by the multiple paths is able to achieve better performance on the data of other clients.

Table 6: Network architecture of FedMP-F. The gray background represents the parameters that need to be communicated.

| Layer / Kernel / Stride | | Output shape | # Params |
|---|---|---|---|
| Path-local | Path-global | - | - |
| ResNet-18 / - / - | ResNet-18 / - / - | 1,000 | 11.7M×2 |
| FC / - / - | | 100 | 0.1M |
| FC / - / - | | 10 / 100 | 1k / 10k |

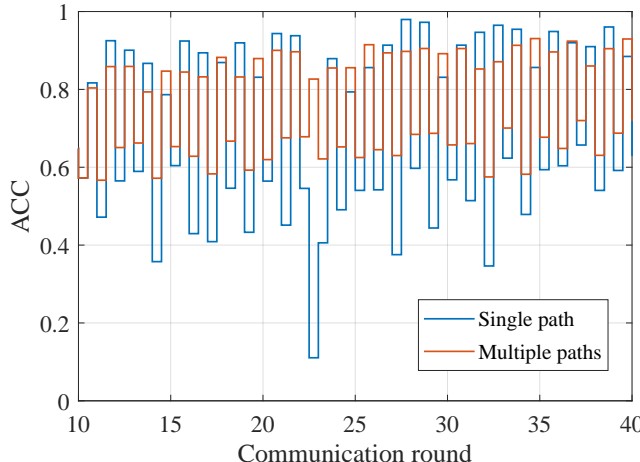

Figure 11: Accuracy curves of the models. The two sides of the horizontal axis ticks are the classification accuracies of the local trained and global aggregated models on the data of current client, respectively. The multiple paths approach achieves less performance degradation and softer fluctuations, with slighter vanishing of local features.

Table 7: The results of CIFARs datasets for all methods. **Bold** and underlined results are the best and the second best.

| Method | CIFAR-10 | | | CIFAR-100 | | |
|---|---|---|---|---|---|---|
| | $\alpha$=0.3 | $\alpha$=0.5 | $\alpha$=1.0 | $\alpha$=0.3 | $\alpha$=0.5 | $\alpha$=1.0 |
| FedAvg | 80.66 | 83.34 | 87.46 | 57.25 | 59.13 | 61.34 |
| FedProx | 81.16 | 83.67 | 88.04 | 56.37 | 57.77 | 62.36 |
| SCAFFOLD | 80.60 | 86.08 | 87.66 | 58.65 | 60.96 | 61.81 |
| FedNova | 83.41 | 84.93 | 86.41 | 55.46 | 55.02 | 56.67 |
| FedDF | 78.70 | 76.95 | 88.65 | 52.82 | 52.46 | 59.32 |
| MOON | 82.47 | 85.46 | 87.83 | 57.52 | 59.30 | 62.85 |
| FedASAM | 68.98 | 71.59 | 74.44 | 46.84 | 47.91 | 48.52 |
| FedPVR | 84.65 | 85.73 | 87.86 | 59.32 | 60.89 | 62.75 |
| FedUCS | 83.61 | 87.79 | 89.43 | 62.19 | 63.24 | 64.28 |
| FedMP | 85.37 | **88.96** | 88.31 | 61.43 | 62.04 | 62.95 |
| FedMP-F | **86.92** | 88.10 | **90.34** | **63.64** | **65.10** | **65.39** |

## E.4 Running Time

The running time of all methods is shown in Figure 12. For the proposed method, full multiple paths solving method is adopted to obtain a fairer comparison. Even when the proposed method adopts the multiple paths solving method, it can still achieve better performance at the cost of a small increase in running time. We believe that this cost is worth it.

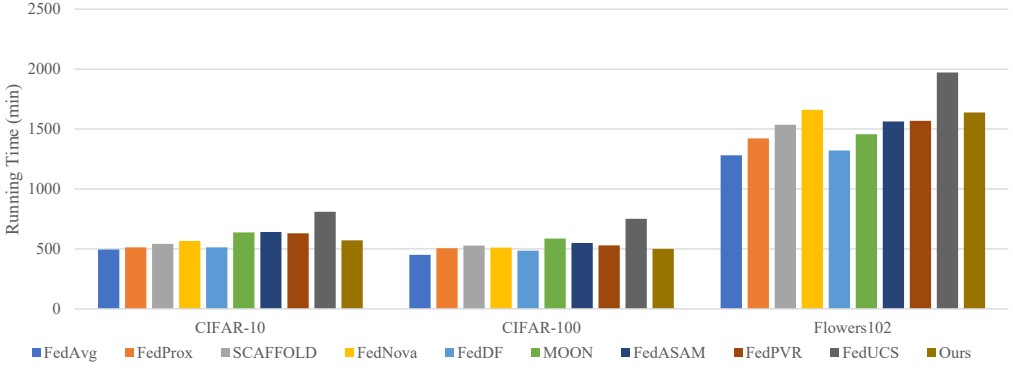

Figure 12: Running time of different methods on three datasets.

