# OpenReview forum: "Soft-consensual Federated Learning for Data Heterogeneity via Multiple Paths"
_NeurIPS.cc/2025/Conference — NeurIPS 2025 poster_

### Official Review · Reviewer_Ky8Q · 2025-06-25

**Clarity:** 3
**Significance:** 3
**Originality:** 3
**Rating:** 4
**Confidence:** 4

**Summary:**

The authors propose a multi-path neural network optimization method from the perspective of global and local soft consensus in federated learning, combining global and local paths to seek their consensus solution. The authors also use swap strategy and data adaptation method to further reduce additional communication overhead. The effectiveness of the proposed method is validated through extensive experiments.

**Questions:**

a. Does the proposed multi-path method require two identical networks to simulate different solution paths? Would this introduce additional network parameters? Could this lead to an unfair comparison when contrasting with other methods?

b. How do you balance global and local paths? Is there any related hyperparameter analysis?

c. What does Figure 9 mean? Please clarify further.

d. What does the binding and swapping in Figure 2 specifically represent? I did not find an explanation for this binding operation in the paper.

**Ethical Concerns:**

["NO or VERY MINOR ethics concerns only"]

**Final Justification:**

Thank you for the authors' reply. Most of my concerns have been addressed, and after reading the other reviewers' comments, I would like to maintain my score.

**Limitations:**

yes

**Quality:**

3

**Strengths And Weaknesses:**

Strengths

a. The idea of using multi-paths to find a consensus solution is insightful.

b. The motivation statement is clear, and the paper provides a detailed explanation of the differences between multi-paths and other similar methods.

c. The paper validates the effectiveness of the proposed framework through a variety of experiments.

Weaknesses

a. The multi-path method appears to introduce additional computational overhead.

b. Balancing different paths introduces additional hyperparameter, and there is no related hyperparameter analysis.

c. The swap strategy may lead to additional information leakage.

---

> ### Author Rebuttal · Authors · 2025-07-30
>
> Thank you for your professional review. Below are our responses to your comments:
>
> * **Response to Weakness (a):** Yes, the multi-path method introduces additional optimization for the extra path. However, we adopt the data adaptation strategy to reduce this additional computational and communication overhead. Moreover, the proposed method improves the performance of federated training and provides a novel perspective for federated learning, which is of exploratory significance. Future work can focus on further reducing the additional computational cost to mitigate this issue.
>
> * **Response to Weakness (b):** Balancing different paths does introduce an additional hyperparameter. However, we can simply set it to 0.1 to avoid complicated hyperparameter tuning. We will include a hyperparameter analysis in the appendix. Below is the hyperparameter analysis result on the CIFAR-100 dataset.
>
> | $\gamma$     | 0.01  | 0.05  | 0.1   | 0.5   | 1     | 5     | 10    | 50    | 100   |
> |--------------|-------|-------|-------|-------|-------|-------|-------|-------|-------|
> | $\alpha$=0.3 | 60.76 | 60.46 | 61.43 | 60.17 | 59.22 | 2.57  | 23.92 | 1.00  | 1.00  |
> | $\alpha$=0.5 | 61.35 | 61.13 | 62.04 | 60.91 | 60.18 | 9.19  | 16.33 | 1.00  | 1.00  |
> | $\alpha$=1.0 | 63.10 | 62.94 | 62.95 | 62.89 | 62.76 | 19.21 | 14.50 | 1.00  | 1.00  |
>
>
> * **Response to Weakness (c):** The swapping strategy indeed carries a potential risk of additional privacy leakage, but this risk is under a controllable level. First, due to the use of data adaptation, the swapping strategy does not require transmitting the full model; thus, the amount of leaked information is limited. Second, since each swap is randomly assigned by the server, it is difficult for clients to infer gradients from the two received models. Furthermore, explicit information such as raw data or label distributions is not directly revealed. Additionally, many existing approaches focus on privacy preservation in federated learning (e.g., differential privacy), and we provide relevant discussions on these methods in the paper.
>
> * **Response to Questions (a):** Yes, the proposed multi-path method requires two networks to simulate two solution paths, which introduces additional parameters during training. However, during model inference, the network architecture and the number of parameters are identical to those of the original model without multi-path learning. Multi-path learning serves only as a training-time enhancement, effectively adding an auxiliary network in parallel to the original network. During training, the two networks optimize from different starting points, and a shared projection ensures consistency in the solution. At inference time, the auxiliary components are discarded, and only the original network is retained, thus no additional parameters are introduced in the final deployed model.
>
> * **Response to Questions (b):** As addressed in our response to weakness (b), we provide the hyperparameter analysis in the appendix. In general, maintaining a balance between the two solution paths is essential for achieving correct multi-path learning outcomes. Moreover, under the swapping strategy, the shared projection is passed along with the local preference path, meaning this path has higher compatibility with the projection. Slightly increasing its weight helps stabilize the multi-path learning process.
>
> * **Response to Questions (c):** Figure 9 shows the model’s accuracy (ACC) on local data before and after local training. Each communication round index on the horizontal axis represents two time points: the time points after model aggregation and after local training. As shown in the figure, compared to single-path learning, multi-path learning requires reaching a soft consensus between paths, making it harder to fit local data—this is reflected in the lower ACC after local training. On the other hand, because the model actively seeks a soft consensus between local and global features during training, more local knowledge is preserved during aggregation across clients. This figure illustrates that multi-path learning achieves a better balance between global and local information in federated learning, leading to an improved global model.
>
> * **Response to Questions (d):** "Binding" means that the shared projection and the main path form a complete and paired solution. During the swapping strategy, this pair of models is assigned together to the same client, hence we refer to them as " Binding." "Swapping" refers to the specific execution step of the swapping strategy.

---

### Official Review · Reviewer_jWBm · 2025-06-26

**Clarity:** 2
**Significance:** 3
**Originality:** 3
**Rating:** 4
**Confidence:** 4

**Summary:**

This paper proposes a federated learning method based on multi-path solution, which seeks a consensus between global feature preferences and local feature preferences through a multi-path approach, avoiding the hard consensus typically achieved by federated learning methods for heterogeneous data. The method balances and jointly optimizes global and local models to construct the aggregated model.

**Questions:**

1. Please respond to what I mentioned in my weaknesses.
2. Could you further clarify the derivations and motivations behind Equations (3)–(7)? While the supplementary material (Section B) effectively highlights the novelty of the multi-path approach, the main text requires more detailed explanations for these Equations.
3. As noted in the weaknesses, there is no discussion on the balance between the two paths. I observed hyperparameters in the multi-path optimization, but the paper lacks a sensitivity analysis on these parameters. Could you address this?
4. In Table 2, the performance appears to degrade when using the multi-path method, which contradicts the conclusions drawn in Figure 3. What explains this discrepancy?

**Ethical Concerns:**

["NO or VERY MINOR ethics concerns only"]

**Limitations:**

Yes

**Quality:**

3

**Strengths And Weaknesses:**

Strengths
1. The paper describes an interesting multi-path solution method. Negotiating between global and local preferences to reach a consensus appears to be an effective solution.
2. The authors provide theoretical proofs in paper.
3. Extensive experiments demonstrate the effectiveness of the proposed method, including comparative studies to showcase the superiority of the multi-path method.

Weaknesses
1. Sections 2.3 and 2.4 contain excessive mathematical notation without a summary table clarifying their meanings.
2. The methodology for data adaptation is not sufficiently elaborated.
3. The discussion lacks an in-depth analysis of the balance between the two paths. Some explanations are unclear, for instance, the relevance of Equations (6) and (7) to the multi-path method is not explain well.

---

> ### Author Rebuttal · Authors · 2025-07-30
>
> Thank you for your professional review. We will respond to each of your comments as follows:
> * **Response to Weakness (1):** We will summarize all symbols used in the paper in a table, which will be added to the appendix. Additionally, each symbol will be explicitly explained at its first appearance in the text.
> | **Symbol** | **Meaning** |
> |--------|--------|
> | $\mathcal{D}_p$ | The dataset of the $p$-th client, which is a subset of the global data |
> | $\|\cdot\|$ | The number of elements in a set |
> | $\hat{\theta}$ | Global model parameters |
> | $\theta_p$ | Model parameters on the $p$-th client |
> | $\mathbf{X}^i_p$ | The $i$-th data sample of the $p$-th client |
> | $\mathcal{L}_p^{(l)}$ | The loss function for the local solution path on the $p$-th client |
> | $\mathcal{L}_p^{(g)}$ | The loss function for the global solution path on the $p$-th client |
> | $\gamma$ | The path balancing hyperparameter, used to balance the influence between local and global paths |
> | $\Phi_p^{(l)}(\cdot)$ | The mapping of the local solution path on the $p$-th client |
> | $\Phi_p^{(g)}(\cdot)$ | The mapping of the global solution path on the $p$-th client |
> | $\Psi_p(\cdot)$ | The shared projection on the $p$-th client |
> | $\phi^r_p$ | The model used by the $p$-th client in the $r$-th communication round to approximate $\Phi^{(l)}_p(\cdot)$ |
> | $\psi_p$ | The model used by the $p$-th client to approximate the projection $\Psi_p(\cdot)$ |
> | $p^{r'}$ | A random sample of client indices that participated in the previous round of training |
> | $\Omega_p(\cdot)$ | The task network on the $p$-th client |
> | $\tau_p^{(l),i}$ | The adapted data generated on the $p$-th client of the $i$-th data sample through the local path mapping |
> | $\tau_p^{(g),i}$ | The adapted data generated on the $p$-th client of the $i$-th data sample through the global path mapping |
>
> * **Response to Weakness (2):** Data adaptation, as an auxiliary strategy of the multi-path method, helps reduce communication overhead. It employs adapters to transform data into a representation space recognizable by the task network, and applies the multi-path method to obtain better solutions for the adapters. In this way, only the adapters need to be exchanged during communication, rather than the entire model.
>
> * **Response to Weakness (3):** We will add a hyperparameter analysis in the appendix. The following table shows the impact of different path balancing hyperparameters on the final performance of the global federated model on the CIFAR-100 dataset. Typically, we set the hyperparameter to 0.1, which aims to balance the two paths as much as possible, with the local preference path serving as the main path and the global preference path as the auxiliary. This is because the shared projection is transmitted along with the local preference path, making it natural to treat the local preference path as the primary one. Equations (6) and (7) describe the initial states of the local preference path models at the beginning of each local training round, which constitutes a formal description of the swapping strategy, indicating that at the start of each local training, the local preference path is initialized with a randomly selected client’s model.
>
> | $\gamma$     | 0.01  | 0.05  | 0.1   | 0.5   | 1     | 5     | 10    | 50    | 100   |
> |--------------|-------|-------|-------|-------|-------|-------|-------|-------|-------|
> | $\alpha$=0.3 | 60.76 | 60.46 | 61.43 | 60.17 | 59.22 | 2.57  | 23.92 | 1.00  | 1.00  |
> | $\alpha$=0.5 | 61.35 | 61.13 | 62.04 | 60.91 | 60.18 | 9.19  | 16.33 | 1.00  | 1.00  |
> | $\alpha$=1.0 | 63.10 | 62.94 | 62.95 | 62.89 | 62.76 | 19.21 | 14.50 | 1.00  | 1.00  |
>
>
> * **Response to Questions (2):** These equations are indeed customized modifications of the multi-path scheme within the federated learning setting. The swapping strategy is a mechanism designed to maintain balance between the two paths in federated learning, preventing one path from becoming overly dominant. Equations (3), (4), and (5) describe the initial state of the model on client p without the swapping strategy, while Equations (6) and (7) describe the initial state on client p when the swapping strategy is applied.
>
> * **Response to Questions (3):** As mentioned in our response to Weakness (3), we have added the experimental results for this hyperparameter analysis.
>
> * **Response to Questions (4):** Indeed, when only the multi-path method is used without the swapping strategy, performance degrades. This precisely highlights the importance of correct multi-path construction for achieving good results. Without the swapping strategy, one path corresponds to a local model trained repeatedly on local data without communication, making it prone to overfitting on that local data. Consequently, during multi-path learning, this model would provide misleading or inappropriate guidance to the global model. Since our goal is to obtain a generalizable model that performs well across all clients, overly strong local bias is harmful to generalization. This is exactly why we adopt the swapping strategy. With swapping, we can still introduce local information that differs from the global model, using a model derived from additional local data as the starting point for one optimization path, thereby enabling a proper multi-path construction. In summary, providing an incorrect path can degrade performance even when using a multi-path framework. Hence, additional strategies are necessary.

---

> ### Comment · Reviewer_jWBm · 2025-08-04
>
> After reading the response, most of my concerns have been solved. I maintain my score.

---

> > ### Author Response · Authors · 2025-08-05
> >
> > Thank you for your professional comments, which have greatly helped improve the quality of the paper.

---

### Official Review · Reviewer_S2r7 · 2025-06-30

**Clarity:** 4
**Significance:** 3
**Originality:** 4
**Rating:** 5
**Confidence:** 5

**Summary:**

To address the data heterogeneity challenge in federated learning, this paper proposes a FedMP framework. This method optimizes multiple solution paths that prefer global and local features and allows them to negotiate to produce a better solution. To balance the path strength, the paper introduces a swapping strategy and uses data adaptation to reduce communication overhead. Comprehensive experiments show that this method outperforms existing advanced methods under heterogeneous data.

**Questions:**

I suggest that the authors provide the fundamental rationale and conceptual clarity of the "local preference swapping strategy", analyze the performance bottlenecks that may be caused by data adaptation, and the choice of loss function. For details, see the previous "Weaknesses".

**Ethical Concerns:**

["NO or VERY MINOR ethics concerns only"]

**Final Justification:**

The rebuttal has addressed my initial concerns. The explanation for the local preference swapping strategy is clear. Besides, the new ablation study addresses my question regarding the performance limitations. These clarifications have resolved my main concerns, and I have raised my score accordingly.

**Limitations:**

yes

**Paper Formatting Concerns:**

This paper complies with the NeurIPS 2025 Paper Formatting Instructions and has no obvious formatting issues.

**Quality:**

3

**Strengths And Weaknesses:**

Strengths
1. The article is well-structured, the problem and motivation are clearly described, and the proposed method is easy to understand.
2. The proposed method is innovative, and it forms a complete solution by coordinating multiple paths with different feature preferences through a shared projection layer, combined with swapping strategies and data adaptation.
3. The authors provide comprehensive experimental results to demonstrate the effectiveness of the method and each module.

Weaknesses
1. In local preference swapping strategy, the local model of the client is not the model trained locally in the previous round, but a local model trained by another client randomly downloaded from the server. Although ablation experiments prove its effectiveness, please explain its rationality, which seems counterintuitive. Besides, the local model of client A may come from client B in the previous round. The paper claims that this path represents "local feature preference". However, for client A, it is actually using the local preference model of client B. Directly describing the preferences of other clients as local preferences may cause ambiguity.
2. In multiple paths data adaptation, to reduce communication costs, FedMP keeps the full task network constant and transfers the optimization objective from a large network to a data adapter. Will this limit the optimal model performance?
3. I would like to know what loss function L_p used in the experiment

---

> ### Author Rebuttal · Authors · 2025-07-30
>
> Thank you very much for your questions and professional comments. Below are our responses:
> * **Response to Weakness (1):** In our experiments, we found that if the swapping strategy is not adopted and the multi-path solving approach is used alone, one of the solution paths would continuously receive local data, leading to overfitting. This significantly degrades the performance of the multi-path method, as demonstrated in our ablation studies: without the swapping strategy to weaken the local-preference path, this path becomes overly dominant and unbalanced, which is an incorrect usage of multi-path learning and results in performance degradation. Therefore, we adopt the swapping strategy to balance the influence of local data on the model. The multi-path method requires two solution paths that optimize from different initial states toward the final models, which can be regarded as two distinct solution paths. If the initial state is close to a suitable value (i.e., derived from actual data rather than random initialization), it reduces the optimization difficulty of the model. Hence, we use the swapping strategy to obtain reasonable initial values from other clients for constructing the multi-path setup. Since this path originates from a model trained on a specific client, we distinguish it from the initial state derived from the global model by referring to it as a local feature preference. This terminology is introduced solely to differentiate the two solution paths. The term "local feature preference" indicates that, unlike the global model, this model has only been exposed to limited data, preserving unique information that has not yet been overwritten or forgotten by other data. This provides a unique interpretation of the current client's data, and this path is expected to reach consensus jointly with the global model, helping to avoid the mediocre solutions that may arise from relying solely on the global model. In summary, the swapping strategy is employed to ensure that, while maintaining multiple solution paths, the starting points of these paths are meaningful and appropriate.
>
> * **Response to Weakness (2):** Yes, it does. Therefore, we provide experimental results without data adaptation, in which the task network is fully optimized using the multi-path method, achieving better performance. Below is a comparison between the full multi-path version and the method described in the main text. The detailed experiment will be updated in the paper.
>
> | Method  | CIFAR-10 ($\alpha=0.3$) | CIFAR-10 ($\alpha=0.5$) | CIFAR-10 ($\alpha=1.0$) | CIFAR-100 ($\alpha=0.3$) | CIFAR-100 ($\alpha=0.5$) | CIFAR-100 ($\alpha=1.0$) |
> |---------|--------------------------|--------------------------|--------------------------|----------------------------|----------------------------|----------------------------|
> | FedMP   | 85.37                    | **88.96**                | 88.31                    | 61.43                      | 62.04                      | 62.95                      |
> | FedMP-F | **86.92**                | 88.10                    | **90.34**                | **63.64**                  | **65.10**                  | **65.39**                  |
>
> * **Response to Weakness (3):** Here, $L_p$ denotes the loss function required for training the task on a client. For the image classification experiments in this paper, we use cross-entropy as the loss function.

---

> ### Comment · Reviewer_S2r7 · 2025-08-03
>
> The response addresses my concern, so I will raise my score.

---

> > ### Author Response · Authors · 2025-08-05
> >
> > Your insightful and professional comments have been instrumental in enhancing the overall quality of the paper, and we are truly grateful.

---

### Official Review · Reviewer_eSzu · 2025-07-02

**Clarity:** 3
**Significance:** 3
**Originality:** 3
**Rating:** 5
**Confidence:** 4

**Summary:**

A multi-path solution method is proposed to combine global and local features in federated learning. This method achieves soft consensus among multiple paths, avoiding the previous approaches of using global information as the sole criterion. Considering the dominance of local features in local training, the authors employ an swapping strategy to weaken the local solution path. They adopt a data-adaptive approach to reduce the amount of model parameter transmission, thereby minimizing additional communication overhead.

**Questions:**

i. Can the proposed method be applied to other models or tasks beyond image classification?

ii. Does the multi-path method employed in the paper enhance the final model's capabilities by increasing the number of model parameters?

iii. How is the data adaptation method described in the paper specifically implemented to reduce model communication overhead?

iv. If two networks are trained together under the same loss function, can this be considered multi-path learning?

v. Is the proposed multi-path learning scheme unrelated to federated learning?

**Ethical Concerns:**

["NO or VERY MINOR ethics concerns only"]

**Final Justification:**

With the improvements provided in their response, my concerns have been adequately addressed.

**Limitations:**

Yes

**Quality:**

3

**Strengths And Weaknesses:**

Strengths

i. The proposed multi-path solution method is interesting, and experiments are conducted to demonstrate its feasibility.

ii. A case study is provided at the beginning to justify the necessity of the proposed method.

iii. The authors explain the multi-path method to distinguish it from other methods.

Weaknesses

i. The validation experiments seem to be limited to image classification, lacking other types of tasks.

ii. The paper lacks hyperparameter analysis, such as analysis of path balancing.

iii. Some descriptions in the paper are unclear and require further refinement.

---

> ### Author Rebuttal · Authors · 2025-07-30
>
> Thank you very much for your professional comments. We will respond to each of your concerns as follows:
> * **Response to Weakness (i):** The goal of our proposed method is to seek a better global model in federated learning, and the use of image data is merely a way to validate the effectiveness of our method under federated settings. A suitable federated approach should not be restricted to a specific data type. Our original intention is to develop a method for obtaining a better global model under federated requirements, and thus we use image data as a case for validation. Similar to image classification, many tasks fundamentally involve learning data embeddings and modeling target probability distributions; hence, our method can be easily extended to tasks on other datasets. We have conducted additional experiments on the Vehicle Sensor dataset, a vehicle dataset containing 23 groups of sensor data, where each sensor group serves as the data for one client. This experiment will be included in the appendix. The following table presents the results of our method compared with several baselines:
> | Method   | FedAvg | FedProx | SCAFFOLD | FedPVR     | FedUCS | FedMP       |
> |----------|--------|---------|----------|------------|--------|-------------|
> | ACC      | 86.33  | 86.12   | 87.14    | 87.71    | 87.53  | **87.93**   |
>
> * **Response to Weakness (ii):** We will add a line plot showing the impact of the path balancing hyperparameter on the performance of the global model. Below is the table for the hyperparameter analysis experiment in CIFAR-100 dataset.
> | $\gamma$     | 0.01  | 0.05  | 0.1   | 0.5   | 1     | 5     | 10    | 50    | 100   |
> |--------------|-------|-------|-------|-------|-------|-------|-------|-------|-------|
> | $\alpha$=0.3 | 60.76 | 60.46 | 61.43 | 60.17 | 59.22 | 2.57  | 23.92 | 1.00  | 1.00  |
> | $\alpha$=0.5 | 61.35 | 61.13 | 62.04 | 60.91 | 60.18 | 9.19  | 16.33 | 1.00  | 1.00  |
> | $\alpha$=1.0 | 63.10 | 62.94 | 62.95 | 62.89 | 62.76 | 19.21 | 14.50 | 1.00  | 1.00  |
>
> * **Response to Weakness (iii):** We will revise or supplement the relevant descriptions based on your feedback.
>
> * **Response to Questions (i):** As mentioned in our response to Weakness(i), the objective of our proposed method is to improve the global model in federated learning, and the use of image data is only one way to verify the effectiveness of our method under federated settings. A proper federated method should not be limited to specific data types. Therefore, we have added an experiment on an additional data type. As shown in the above table, our method still achieves a certain level of performance improvement.
>
> * **Response to Questions (ii):** Here we would like to clarify that, for the multi-path method, the inference structure of the original model does not need to be changed. Therefore, during the final testing phase, the model used has the same architecture as the original model, and thus the number of parameters remains unchanged. The multi-path method only requires parallel auxiliary network structures during training to simulate different solution paths, which introduces additional auxiliary parameters. However, after training is completed, these auxiliary parameters are no longer involved in the model inference process. Hence, the improvement in federated model performance is not due to an increase in the number of parameters.
>
> * **Response to Questions (iii):** We connect an adapter between the task network, and the data to transform the raw data into a format suitable for the task network. For multi-path solving, we also attach a shared projection adapter after the task network to convert its output into the desired target output. In this case, the multi-path solving process is transformed from optimizing the entire model to optimizing only the adapters. As a result, only the adapter parameters need to be communicated, reducing communication overhead.
>
> * **Response to Questions (iv):** We believe the scenario you mentioned does not qualify as multi-path learning. The key of multi-path learning lies in having some distinct paths that negotiate and reach a consensus solution. Simply training under the same loss function without generating a consensus solution does not constitute multi-path learning. In our proposed multi-path construction, two parallel networks feed into a shared projection module, which maps their outputs into the target space (i.e., the desired output). This projection can be viewed as a consensus reached through the collaboration of the two parallel networks, with each network representing a separate solution path. Only approaches that involve such a negotiation process can be considered the multi-path learning.
> * **Response to Questions (v):** The multi-path scheme is a general idea for finding consensus between two solution paths. How to concretely implement such a multi-path solution within the federated learning framework requires further exploration. In this paper, we combine a global-preference path with a local-preference path to achieve a soft consensus, which addresses the conflict between global and local knowledges in federated learning. Therefore, this constitutes a federated learning method based on multi-path learning.

---

> > ### Comment · Reviewer_eSzu · 2025-08-04
> > **Some questions**
> >
> > Thank you for your response, which has addressed my concerns. However, after reviewing the other reviewers' comments, I still have some additional questions:
> >
> > a. The authors mention that their method provides two distinct solution paths and employs a multi-path optimization approach to obtain a consensus solution between these paths for federated model training. Does this solution bear some similarity to mixture-of-experts models? The authors need to further clarify this point.
> >
> > b. Have the authors made any attempts to incorporate privacy-preserving methods into their proposed approach?

---

> > > ### Author Response · Authors · 2025-08-05
> > >
> > > Thank you for your previous professional comments, which have greatly helped improve the quality of this paper. Below are my responses to the new questions:
> > >
> > > * A. I understand your confusion regarding the fact that both Mixture-of-Experts (MoE) models and multi-path methods involve multiple data pathways. However, the goal of multi-path solving is to reach a consensus solution from two distinct solution paths, which is fundamentally different from MoE. Unlike MoE, multi-path solving focuses on achieving consensus based on the original model structure. In essence, multi-path solving aims to produce a better solution *within* the original model, whereas MoE leverages a larger number of network components to fit complex and diverse tasks. Therefore, the focus and objectives of these two techniques are quite different. Multi-path solving emphasizes negotiation toward a global consensus, softly integrating diverse knowledge *within a single model* during training. In contrast, MoE emphasizes preserving personalized knowledge by using multiple specialized models to store distinct knowledge patterns.
> > >
> > > * B. The proposed method is designed to build federated models under data heterogeneity, with a primary focus on mitigating the negative effects of non-IID data. Regarding privacy-preserving methods, numerous researchers have already proposed relevant approaches. As a federated learning method aimed at addressing data heterogeneity, FedMP, like FedAvg, can be extended to incorporate additional privacy-preserving mechanisms. As an exploratory validation, we integrate differential privacy (DP) into FedMP to meet higher privacy requirements, and conduct a preliminary comparison with FedAvg enhanced with the same DP mechanism. We use a backbone network consisting of two CNN layers and two fully connected layers, and apply DP with a noise multiplier of 1.0 on the CIFAR-10 dataset. The results are shown in the following table.
> > >
> > > | Method         | $\alpha=0.3$ | $\alpha=0.5$ | $\alpha=1.0$ |
> > > |----------------|--------------|--------------|--------------|
> > > | FedAvg         | 61.85        | 61.89        | 62.19        |
> > > | FedAvg-DP | 61.13        | 61.07        | 60.73        |
> > > | FedMP          | 63.31        | 63.09        | 65.16        |
> > > | FedMP-DP  | 62.47        | 62.34        | 63.58        |
> > >
> > > As the table indicates, even with the addition of privacy-preserving mechanisms, the proposed method still maintains a relative performance advantage. This demonstrates that FedMP can be extended with such strategies to meet stronger privacy protection requirements.

---

> > > > ### Comment · Reviewer_eSzu · 2025-08-08
> > > > **Thanks**
> > > >
> > > > Thank you for your thoughtful response and efforts. My concerns have been adequately addressed, and I will update my score accordingly.

---

### Decision · Program_Chairs · 2025-09-17

**Decision:**

Accept (poster)

**Comment:**

This paper proposes a multi-path soft consensus learning framework for federated learning, which seeks consensus between global and local features to generate a better federated model. The authors conduct extensive experiments to validate the effectiveness of the multi-path approach. During the rebuttal phase, the authors provide responses to reviewers' concerns regarding path balancing and privacy issues. After the rebuttal, the reviewers' evaluations remain positive. Hence, I recommend its acceptance.